# Machine learning improves seasonal mass balance prediction for unmonitored glaciers

Kamilla Hauknes Sjursen<sup>1</sup>, Jordi Bolibar<sup>2,3</sup>, Marijn van der Meer<sup>4,5</sup>, Liss Marie Andreassen<sup>6</sup>, Julian Peter Biesheuvel<sup>3</sup>, Thorben Dunse<sup>1</sup>, Matthias Huss<sup>4,5,7</sup>, Fabien Maussion<sup>8,9</sup>, David R. Rounce<sup>10</sup>, and Brandon Tober<sup>10</sup>

**Correspondence:** Kamilla Hauknes Sjursen (kasj@hvl.no)

Abstract. Glacier evolution models based on temperature-index approaches are commonly used to assess hydrological impacts of glacier changes. However, current model calibration frameworks cannot efficiently transfer information from sparse high-resolution observations across glaciers. This limits their ability to resolve seasonal mass changes on unmonitored glaciers in large-scale applications. Machine learning approaches can potentially address this limitation by learning relationships from sparse data that are transferable in space and time, including to unmonitored glaciers. Here, we present the Mass Balance Machine (MBM), a data-driven mass balance model based on the XGBoost architecture, designed to provide accurate and high spatio-temporal resolution regional-scale reconstructions of glacier mass balance. We trained and tested MBM using a dataset of approximately 4000 seasonal and annual point mass balance measurements from 32 glaciers across heterogeneous climate settings in mainland Norway, spanning from 1962 to 2021. To assess the advantage of MBM's generalisation capabilities, we compared its predictions on independent test glaciers at various spatio-temporal scales with those of regional-scale simulations from three glacier evolution models. MBM successfully predicted annual and seasonal point mass balance on the test glaciers (RMSE of 0.59-1.00 m w.e. and bias of -0.01-0.04 m w.e.). On seasonal mass balance, MBM outperformed the other models across spatial scales, reducing RMSE by up to 46% and 25% on glacier-wide winter and summer mass balance, respectively. Our results demonstrate the capability of machine learning models to generalise across glaciers and climatic settings from relatively sparse mass balance data, highlighting their potential for a wide range of applications.

<sup>&</sup>lt;sup>1</sup>Department of Civil Engineering and Environmental Sciences, Western Norway University of Applied Sciences (HVL), Sogndal, Norway

<sup>&</sup>lt;sup>2</sup>Univ. Grenoble Alpes, CNRS, IRD, G-INP, Institut des Géosciences de l'Environnement, Grenoble, France

<sup>&</sup>lt;sup>3</sup>Faculty of Civil Engineering and Geosciences, Delft University of Technology, Delft, The Netherlands

<sup>&</sup>lt;sup>4</sup>Laboratory of Hydraulics, Hydrology, and Glaciology (VAW), ETH Zürich, Zurich, Switzerland

<sup>&</sup>lt;sup>5</sup>Swiss Federal Institute for Forest, Snow, and Landscape Research (WSL), Sion, Switzerland

<sup>&</sup>lt;sup>6</sup>Norwegian Water Resources and Energy Directorate (NVE), Oslo, Norway

<sup>&</sup>lt;sup>7</sup>Department of Geosciences, University of Fribourg, Fribourg, Switzerland

<sup>&</sup>lt;sup>8</sup>Bristol Glaciology Centre, School of Geographical Sciences, University of Bristol, Bristol, UK

<sup>&</sup>lt;sup>9</sup>Department of Atmospheric and Cryospheric Sciences, University of Innsbruck, Innsbruck, Austria

<sup>&</sup>lt;sup>10</sup>Department of Civil and Environmental Engineering, Carnegie Mellon University, Pittsburgh, PA, USA

#### 1 Introduction

Glaciers around the world are losing mass and retreating due to atmospheric warming (IPCC, 2019), with numerous impacts on nature and society (Schaub et al., 2013; Huss et al., 2017; Milner et al., 2017; Varnajot and Saarinen, 2021; Emmer et al., 2022; Bosson et al., 2023). Glaciers represent significant freshwater reservoirs that modulate downstream freshwater availability throughout the year. Climate change alters the timing and magnitude of glacier runoff (Bliss et al., 2014; Huss and Hock, 2018; Wimberly et al., 2024), which subsequently affects the hydrology of glacierised catchments (Nie et al., 2021; Ultee et al., 2022). The influence of climatic forcing is reflected in the glacier surface mass balance, which refers to the change in mass at the surface of a glacier, or a part of a glacier, over a given period (Cogley et al., 2011), usually a year (annual mass balance) or a season (winter or summer mass balance for mid-latitude glaciers). Quantifying glacier runoff requires reliable mass balance estimates at high spatio-temporal resolution (i.e. individual glaciers, monthly to seasonal estimates). Such detailed assessments are essential for societies to adapt effectively to the impacts of climate change on the hydrological system.

Glacier mass balance models are valuable tools for quantifying glacier mass changes (Radić and Hock, 2011; Bliss et al., 2014; Huss and Hock, 2015, 2018; Marzeion et al., 2012; Maussion et al., 2019; Shannon et al., 2019; Zekollari et al., 2019; Rounce et al., 2023). Most large-scale glacier evolution models estimate glacier mass balance using temperature-index approaches that parametrise the relationship between surface melt and temperature (Hock et al., 2019; Marzeion et al., 2020). To ensure that mass changes and climate sensitivities are accurately captured at the scale of individual glaciers, model parameters (e.g. precipitation bias-correction factors and degree-day factors that relate the amount of ice, snow, or firn melt to temperature) must be calibrated using glacier-specific mass balance observations (Rounce et al., 2020b; Schuster et al., 2023; Zekollari et al., 2024). Traditionally, such observations have been limited to in situ surveys using the glaciological method, where surface mass balance measurements are performed at a network of mass balance stakes (point mass balance), and seasonal and annual components are interpolated over the glacier area (glacier-wide mass balance; Østrem and Brugman, 1991). However, since in situ mass balance observations are challenging and resource intensive, their availability is extremely limited on a global scale (around 0.02% of the worlds glaciers; WGMS, 2023). The scarcity of glacier-specific observations has historically posed a major challenge in calibrating temperature-index approaches (e.g. Radić and Hock, 2014). Significant efforts have been made to develop suitable calibration techniques using limited data (e.g. Radić and Hock, 2011; Huss and Hock, 2015). However, large-scale models still suffer from transferability issues: they lack efficient frameworks to leverage sparse in situ observations for quantifying mass changes on unmonitored glaciers.

The increasing availability of geodetic mass balance observations has recently alleviated the lack of glacier-specific observations. These observations assess glacier surface elevation changes from time series of satellite-derived digital elevation models (DEMs) over decadal time scales (e.g. Dussaillant et al., 2019; Shean et al., 2020; Hugonnet et al., 2021). Most large-scale glacier evolution models today perform glacier-specific parameter calibration using (multi-year) satellite-derived geodetic mass balance observations (e.g. Rounce et al., 2020a, 2023; Caro et al., 2024; Kang et al., 2024; Zekollari et al., 2024) due to their worldwide coverage (2000–2019; Hugonnet et al., 2021). However, satellite-derived geodetic observations do not provide sufficient constraints to quantify seasonal mass changes (Rounce et al., 2020b; Sjursen et al., 2023). This results in *equifinality* 

50 (Beven, 2006); multiple parameter sets, and thus combinations of accumulation and melt, can accurately reproduce the observed net mass changes. Consequently, the sparsity of in situ seasonal observations and transferability issues facing current large-scale glacier evolution models still hamper their ability to produce reliable estimates of seasonal runoff magnitudes for unmonitored glaciers.

In recent years, the use of machine learning (ML) to model glacier mass balance has emerged as a promising approach to address some of the limitations of temperature-index approaches (Steiner et al., 2005; Bolibar et al., 2020, 2022; Anilkumar et al., 2023; Guidicelli et al., 2023; Diaconu and Gottschling, 2024; van der Meer et al., 2025). ML models generalise patterns from training data and apply them to make accurate inferences on new, independent data. They can thus learn statistical relationships between mass balance components and topographical and meteorological variables that are transferable across space and time, including to unsurveyed glaciers and years (e.g. Guidicelli et al., 2023). This means that ML models can leverage sparse in situ data, such as annual and seasonal glaciological measurements, to provide high spatio-temporal resolution mass balance estimates of unmonitored glaciers across a larger region. They thus have the potential to improve the accuracy of such predictions, compared to temperature-index approaches that rely on glacier-specific calibration to multi-year geodetic observations.

A promising avenue of ML in large-scale glacier mass balance modelling is the spatio-temporal generalisation of highresolution information from seasonal and annual mass balance observations at the point scale. Point mass balance measurements from glaciological surveys represent the most direct and high-quality observations of glacier surface mass balance. These measurements offer a level of precision that surpasses glacier-wide surface mass balance, which relies on inter- and extrapolating data to unmeasured areas. To our knowledge, only two studies have trained ML models on point mass balance measurements, van der Meer et al. (2025) presented an ML approach for reconstruction of annual mass balance at specific sites on glaciers in Switzerland by training one model for each individual stake using temporal aggregations of temperature and precipitation as input features. Anilkumar et al. (2023) used annual point mass balance data from glaciers in the European Alps to compare the performance of various ML architectures. However, their use of random train-test splits on spatially correlated data means that a robust assessment of the ability of ML models to generalise across different locations is still lacking. Moreover, the use of seasonal data is still largely unexplored and limited to using elevation-band winter mass balance to assess precipitation biases in climate reanalysis products (Guidicelli et al., 2023). Generalising from seasonal and annual point mass balance measurements offers the potential to provide high temporal resolution distributed mass balance predictions on unmonitored glaciers, ultimately improving runoff predictions from glacierised catchments. Moreover, the advantages and limitations of ML methods in this context compared to traditional modelling approaches remain unclear. Such a comparison would clarify how ML-based mass balance models could serve as a useful and complementary tool to enhance the accuracy of glacier mass balance predictions.

This study aims to evaluate the ability of an ML model to generalise spatio-temporal information across glaciers using seasonal and annual point mass balance measurements, with the goal of providing accurate, high-resolution predictions of surface mass balance on unmonitored glaciers in regional-scale applications. We present the Mass Balance Machine (MBM), a data-driven mass balance model based on eXtreme Gradient Boosting (XGBoost; Chen and Guestrin, 2016), capable of re-

constructing surface mass balance up to a point scale and monthly temporal resolution for independent glaciers with diverse configurations and climatic settings across Norway. Herein, we demonstrate how MBM can incorporate observations at different temporal scales (seasonal and annual) in training and be customised to generate predictions at an even finer (monthly) temporal resolution. To assess the potential of MBM to improve glacier mass balance estimates on unmonitored glaciers, we compare its performance with state-of-the-art large-scale glacier evolution models that rely on temperature index approaches to estimate melt and are calibrated using existing frameworks and satellite-derived geodetic mass balance: the Global Glacier Evolution Model (GloGEM; Huss and Hock, 2015), the Open Global Glacier Model (OGGM; Maussion et al., 2019) and the Python Glacier Evolution Model (PyGEM; Rounce et al., 2023). Modelled mass balances are compared to observations at point to glacier-wide spatial scales and seasonal-to-decadal scale resolution. In addition, we benchmark monthly predictions across all models (without observations). In light of our findings, we discuss the potential applications of MBM, as well as future perspectives on ML-based mass balance models.

#### 2 Mass balance dataset and study area

We used a dataset of annual and seasonal glaciological point mass balance measurements from glaciers in mainland Norway (Elvehøy et al. (2025); Fig. 1), collected by the Norwegian Water Resources and Energy Directorate (NVE) (e.g. Kjøllmoen et al., 2024), to train MBM. The dataset contains measurements at 4170 stakes (unique combinations of locations and years) on 32 individual glaciers on the Norwegian mainland (3082/1088 stakes on 22/10 glaciers in southern/northern Norway; Fig. 1). Each of the 4170 stakes has between one and three readings (annual, summer and/or winter), totalling 3910 annual, 3929 summer and 3751 winter point mass balance measurements (over the period 1962–2021; Fig. 2). In all, the 32 glaciers correspond to an area of 343 km², or ~15% of the total glacierised area in Norway (2328 km² in the 2018/19 glacier inventory; Andreassen et al., 2022). Norwegian glaciers provide a good case study for our investigation due to their diverse characteristics and heterogeneous climatic settings. In southern Norway, glaciers exhibit a strong longitudinal gradient in mass turnover with the transition from the maritime climate of the west coast to the drier interior mountain ranges (Andreassen et al., 2005). Glaciers in northern Norway have a lower equilibrium line altitude, reflecting the increasing latitude toward the Arctic. Measurements in this region also reveal a decrease in mass turnover with distance to the coast, but within a smaller range of values compared to southern Norway.

The dataset constitutes a good representation of the spatio-temporal variability in the characteristics of glaciers in Norway. It includes observations from the main climatic settings (Fig. 1), from the maritime glaciers of northern Norway (e.g. Langfjordjøkelen at 70°10′N, 21°45′E and Engabreen at 66°40′N, 13°45′E), to glaciers along the west-east maritime to continental climate gradient in southern Norway (e.g. Ålfotbreen at 61°45′N, 5°40′E to Gråsubreen at 61°39′N, 8°37′E). A wide elevation range is covered (minimum 190 m a.s.l. to maximum 2212 m a.s.l.; Fig. 2c) reflecting that in northern Norway, the lowest glacier tongues extend almost to sea level, while the highest altitude glaciers in southern Norway reside above 2000 m a.s.l. The dataset has a continuous coverage of the time period 1962–2021 (Fig. 2b). During this period, glaciological records show temporal variations in mass balance, with periods of mass gain and loss (Andreassen et al., 2005, 2020).

Figure 1. Geographical distribution of in situ point mass balance observations in the dataset. Climatic regions are indicated by coloured circles on the main map: purple for northern Norway (N) including Finnmark (FIN), Skjomen (SKJ), Blåmannsisen (BLA) and Svartisen (SVA), blue for the most maritime glaciers in western Norway (W-MAR) in the Ålfotbreen (ALF) region, orange for western Norway (W) including Jostedalsbreen and Breheimen (JOB), Hardangerjøkulen (HAR) and Folgefonna (FOL) and green for the easternmost glaciers in Jotunheimen (JOT) in southern Norway (E). The size of the circles indicates the number of stakes in the dataset for each region. Insets (a)–(e) show location of glaciers used for training (black points) and test (red points). Glacierised areas are shown in dark grey.

#### 3 The Mass Balance Machine (MBM)

The goal of MBM is to predict surface mass balance on glaciers in Norway at high spatio-temporal resolution, based on established relationships between mass balance and glacier characteristics and climatic forcing. This section introduces 1) the chosen ML approach (architecture) for MBM, 2) selection of relevant features (predictors used by MBM) and 3) our strategy for training and testing of MBM, including the design of an independent test dataset for the final performance evaluation of the trained model.

#### 3.1 Architecture

MBM is based on the gradient-boosted ensemble decision tree-based method XGBoost (Chen and Guestrin, 2016). Decision trees resemble tree-like structures similar to flowcharts, with nodes, branches and leaves representing possible decision points,

**Figure 2.** Characteristics of the point mass balance dataset, in terms of (a) distribution of observations of annual (black), winter (blue) and summer (red) mass balance, (b) number of observations per year, and distribution of topographical features associated with each stake: (c) elevation, (d) aspect and (e) slope.

partitions and outcomes (numerical targets). A well-known problem with decision tree learners is that they can result in overcomplex trees that tend to overfit, i.e. they do not generalise well beyond their training domain. Ensemble models attempt to overcome this issue by building a strong learner from an ensemble of weak learners (relatively simple trees), which both reduces bias and variance in predictions. XGBoost is based on the ensemble method of boosting, where weak learners are trained iteratively with the goal of developing new trees that improve predictions of previous trees. In XGBoost, the sequential development of weak learners is based on reducing the model error by fitting the residuals of the current ensemble of trees.

On medium-sized (

**Figure 3.** Features used by the Mass Balance Machine. Meteorological features are retrieved from the ERA5-Land grid cell closest to the location of a point surface mass balance (SMB) measurement, while topographical features are extracted from the nearest Digital Elevation Model (DEM) grid cell (Copernicus DEM GLO-90 at 90 m resolution). The resolution of the ERA5-Land and DEM grid cells in the figure are not to scale.

trained model on the test dataset. Step 3 serves as an assessment of the predictive power of the model on new unseen data. In the following sections, we detail our strategies for the design of the independent test dataset, tuning of MBM hyperparameters and performance evaluation.

#### 3.3.1 Test dataset

190

The test dataset used for ML model performance evaluation must be carefully selected with respect to the modelling objective, such that the model's performance on the test dataset accurately reflects its ability to fulfil this objective. In addition, an underlying assumption of this performance evaluation is that the data used for model training and testing are independent (e.g. Hastie et al., 2009, Chapter 7). Point mass balance data, however, exhibit spatial correlation, meaning that measurements taken at one location will be similar to those at nearby locations. Random train-test data splits, which are commonly used in ML, do not ensure independence between the training and test datasets in the presence of spatial correlation. In such cases, a random split may result in leakage of information between training and test datasets, leading to unreliable performance measures that may not be representative of the model's true performance on independent data (e.g. Roberts et al., 2017; Schratz et al., 2019; Kattenborn et al., 2022). Thus, when independence may be compromised, as with spatially correlated point mass balance measurements, the train-test split should be designed to minimise autocorrelation between training and test data, for example, by using spatial blocking strategies (Roberts et al., 2017).

**Figure 4.** Illustration of the set-up and training of MBM showing (a) the structure of the tabular dataset with features and targets for each seasonal or annual point mass balance measurement (ID), (b) monthly restructuring of features to facilitate monthly predictions for each ID, and (c) aggregation of monthly predictions to seasonal and annual values for evaluation (MSE loss) against the corresponding targets.

Considering these criteria, all point mass balance measurements for 14 glaciers in the dataset were withheld for testing. The test glaciers were chosen from each region in Fig. 1, such that the distributions of targets and features in the test dataset are similar to those of the training dataset. We avoided selecting adjacent glaciers to ensure independence between the training and test data to the best of our ability. The performance evaluation of MBM on the test dataset thus reflects the model's ability to predict mass balance on glaciers without mass balance observations. We consider the test glaciers to be representative of the population of Norwegian glaciers, both in terms of climatic settings, topography and mass balance distributions. In total, 1065, 999 and 1028 annual, winter and summer mass balance measurements were withheld for testing, respectively (corresponding to 27, 27 and 26 percent of the total number of measurements).

## 3.3.2 Model training and hyperparameter tuning

During training, MBM learns the structure of the training data by iteratively building trees to minimise a loss function. We employed the commonly used Mean Squared Error (MSE):

205 
$$MSE = \frac{1}{n} \sum_{i=1}^{n} (b_i - \hat{b}_i)^2,$$
 (1)

where  $b_i$  is the target point mass balance and  $\hat{b}_i$  is the predicted point mass balance corresponding to each of the n targets. Before evaluating the MSE loss, monthly point mass balance predictions from MBM are aggregated over the time period associated with each target point mass balance (seasonal or annual, Fig. 4). Thus, a seasonal or annual point mass balance

**Table 1.** Overview of hyperparameter combinations used in Mass Balance Machine hyperparameter tuning and selected hyperparameter combination during cross-validation. n\_estimators refers to the number of trees, max\_depth refers to the maximum tree depth and min\_child\_weight refers to the minimum number of samples required to split a node.

| Hyperparameter   | Grid search                            | Selected |
|------------------|----------------------------------------|----------|
| learning_rate    | [0.01, 0.05, 0.10, 0.15, 0.20]         | 0.05     |
| n_estimators     | $\left[100, 200, 300, 400, 500\right]$ | 300      |
| max_depth        | [3,4,5,6,7]                            | 5        |
| min_child_weight | [0, 5, 10]                             | 0        |

prediction  $\hat{b}_i$  is the sum of m monthly predictions  $\hat{b}_t$ :

210 
$$\hat{b}_i = \sum_{t=1}^m \hat{b}_t,$$
 (2)

where m equals 12, 7 and 5 for an annual, winter and summer point mass balance prediction, respectively (see Sect. 3.2).

Hyperparameters refer to the parameters of an ML model that are configured before training and control the learning process. We performed a hyperparameter grid search using five-fold cross-validation to identify the optimal hyperparameter configuration (e.g. Hastie et al., 2009, Chapter 7). This involved splitting the training dataset (the remaining training data after the train-test split) into five subsets (folds), repeatedly training the model on four of the five folds, and evaluating the model performance (validation) on the remaining fold. Thus, all folds are used once for validation for each hyperparameter combination. Ideally, the split of training and validation subsets would follow the same spatial blocking strategy as the train/test split to ensure that the choice of model hyperparameters is based on a validation performance that reflects the expected test performance. However, this is not always feasible within the limitations of the data (Roberts et al., 2017). The point mass balance dataset includes a limited number of glaciers. Using a strict spatial blocking strategy in the train/validation split would create unbalanced folds. This introduces unnecessary high demands of the model to extrapolate and hampers learning. As a compromise, we assigned every fifth mass-balance year in the training dataset to a different fold. This train/validation split strategy yields balanced folds (between 518 and 624 annual, winter and summer mass balance measurements per fold), which facilitates learning while forcing the model to perform some extrapolation in time. Importantly, the strategy avoids a random split that would give unreliable validation scores.

During the hyperparameter grid search, different combinations of four XGBoost hyperparameters were used (Table 1): the learning rate, number of estimators, maximum tree depth and the minimum number of samples required to split a node. Other hyperparameter configurations were also investigated but did not have notable effects on the model validation performance. Therefore, the remaining hyperparameters were kept to default values. We selected the hyperparameter combination that minimised the mean MSE of the five validation folds (Table 1).

## 3.3.3 Model performance evaluation on test dataset

Once the optimal model hyperparameters were chosen, MBM was retrained on the full training dataset. Then, we assessed the performance of MBM on the test dataset of annual and seasonal point mass balance measurements described in Sect. 3.3.1 (1065, 999 and 1028 annual, winter and summer point mass balance measurements on 14 glaciers between 1962–2021). For the performance evaluation, MBM's monthly predictions were aggregated to seasonal or annual resolution as done in training (Eq. 2 and Fig. 4c). We assessed the performance of MBM using the following metrics: Root Mean Squared Error (RMSE), Mean Absolute Error (MAE), mean bias and R<sup>2</sup> metric.

## 4 Mass balance model comparison

#### 4.1 Model comparison set-up





MBM predictions were compared at different spatio-temporal scales against those of established global glacier evolution models using temperature-index approaches: GloGEM (Huss and Hock, 2015, with minor updates), OGGM (Maussion et al., 2019) and PyGEM (Rounce et al., 2020b, 2023) (henceforth referred to as glacier evolution models). There are two objectives to this comparison: 1) to compare the performance of MBM to the glacier evolution models using available glacier mass balance observations (glaciological point and glacier-wide mass balance observations (Kjøllmoen et al., 2024), as well as geodetic mass balance observations from different sources (Andreassen et al., 2016, 2020; Hugonnet et al., 2021)) and 2) to benchmark MBM's monthly mass balance predictions against those of the glacier evolution models since no mass balance observations are available on this time scale. We thus compared predictions from MBM and the glacier evolution models across a wide range of spatial scales, from the point/elevation-band scale, over mass balance gradients with elevation to the glacier wide scale, as well as temporal scales, from monthly, seasonal and annual to decadal time periods.

The model comparison was conducted on the same glaciers as in MBM's test dataset (Sect. 3.3.1), but using available observations for the common modelling period 1980–2019 (Table 2). Over this period, glaciological observations (point and glacier-wide) are available for 11 of the 14 glaciers in the test dataset (three glaciers only have measurements from the 1960s and 70s). We chose to compare the models on the test glaciers since they provide independent and rigorous performance measures for MBM, i.e. no data from these glaciers have been used for MBM training. For all models, annual, winter and summer mass balance predictions were computed by aggregating monthly mass balances over the hydrological year (Oct–Sept), winter months (Oct–Apr) and summer months (May–Sept), respectively. With regards to comparison with point measurements, it should be noted that the glacier evolution models provide mass balance averaged over elevation bands rather than in grid cells. GloGEM and OGGM have fixed elevation band intervals, and we extracted the modelled mass balance from the elevation band corresponding to the elevation of each point mass balance measurement. PyGEM results are given in fixed distances along the glacier flowline such that steeper parts of the glacier cover a wider elevation interval.

**Table 2.** Overview of the set-up of glacier evolution model simulations used for model intercomparison. All models use RGI 6.0 and constant area over the simulation period. Calibration is performed for each glacier individually, using geodetic mass balance for 2000–2019 from Hugonnet et al. (2021). The time period refers to mass-balance years (Oct–Sept) covered by the simulations. DDFs refers to degree day factors, and  $P_{corr}$  and  $T_{corr}$  refer to precipitation and temperature bias correction, respectively.

| Model  | Time<br>period | Spatial resolution | Climate           | Temperature downscaling                                       | Precipitation downscaling                  | DDFs                                          | Parameters calibrated                                                                     | Reference                                       |
|--------|----------------|--------------------|-------------------|---------------------------------------------------------------|--------------------------------------------|-----------------------------------------------|-------------------------------------------------------------------------------------------|-------------------------------------------------|
| GloGEM | 1980–<br>2019  | 10 m <sup>a</sup>  | ERA5 <sup>c</sup> | Monthly lapse rate for each reanalysis grid cell <sup>d</sup> | Vertical gradient $0.025\% \text{ m}^{-1}$ | Separate DDF for snow, ice, firn <sup>e</sup> | $P_{\mathrm{corr}},$ $T_{\mathrm{corr}}{}^f,$ $DDF_{\mathrm{snow}},$ $DDF_{\mathrm{ice}}$ | Huss and Hock (2015)                            |
| OGGM   | 1961–<br>2019  | 30 m <sup>a</sup>  | W5E5 <sup>c</sup> | Lapse rate<br>6.5 K km <sup>-1</sup>                          | None                                       | Single DDF                                    | $\begin{aligned} &P_{\mathrm{corr}}, T_{\mathrm{corr}}, \\ &DDF \end{aligned}$            | Maussion et al. (2019); Zekollari et al. (2024) |
| PyGEM  | 1961–<br>2022  | 30 m <sup>b</sup>  | ERA5 <sup>c</sup> | Monthly lapse rate for each elevation $bin^d$                 | Vertical gradient $0.01\% \text{ m}^{-1}$  | Separate DDF for snow, ice, $firn^e$          | $P_{corr}, T_{corr}, \\ DDF_{snow}{}^g$                                                   | Rounce et al. (2023)                            |

<sup>&</sup>lt;sup>a</sup> Vertical resolution, mass balance is provided in elevation bands. <sup>b</sup> Horizontal resolution, mass balance is provided along flowlines. <sup>c</sup>Monthly temperature and precipitation.

#### 4.2 Glacier evolution models

The three glacier evolution models were all run for RGI region 8 (Scandinavia) using RGI 6.0 outlines and a constant glacier area (no glacier dynamics). The RGI 6.0 outlines for Norway are derived from 1999–2006 satellite imagery (Andreassen et al., 2012). Mass balance was predicted on a monthly time scale and for bins along flowlines or in elevation bands, using between one and three degree-day factors to simulate melt on the glacier surface (Table 2). As climate forcing, GloGEM and PyGEM uses ERA5 (Hersbach et al., 2020), while OGGM uses W5E5 (bias-corrected ERA5 over land; Lange et al., 2021). Parameter values were calibrated at the individual glacier scale using geodetic mass balance for the period 2000–2019 (Hugonnet et al., 2021). Depending on the model, three to four free parameters were calibrated. The set-up of each model is detailed in Table 2.

# 4.3 Glacier-wide predictions using MBM

Glacier-wide predictions were produced for MBM with the same set of features as described in Sect. 3.2. For each test glacier, a DEM (Copernicus DEM GLO-90, ~90 m resolution) and RGI 6.0 outline were retrieved using the OGGM pipeline. Then, for each DEM grid cell, monthly meteorological features were obtained from the nearest ERA5-Land cell for 1962–2021. MBM was then run for each glacier to predict the monthly mass balance in every DEM grid cell over the whole time period based

<sup>&</sup>lt;sup>d</sup>Derived from ERA5 pressure levels. <sup>e</sup>DDF<sub>firn</sub> is average of DDF<sub>snow</sub> and DDF<sub>ice</sub>. <sup>f</sup> Only included if no match is found with other parameters within predefined bounds.

<sup>&</sup>lt;sup>g</sup>DDF<sub>ice</sub> set to 0.7DDF<sub>snow</sub>.

on the topographical and monthly meteorological features. Glacier-wide monthly mass balance predictions were produced by intersecting the DEM with the RGI 6.0 glacier outline and aggregating predictions over the glacierised area.

## 5 Results





In this section, we present the performance evaluation of MBM on the test dataset and the comparison of MBM and the glacier evolution models (GloGEM, OGGM and PyGEM). In Sect. 5.1 we focus on the performance of MBM on the full test dataset of seasonal and annual point mass balance measurements (14 glaciers, 1962–2021) described in Sect. 3.3.1. Section 5.2 compares the performance of MBM and glacier evolution models at various spatio-temporal scales using available glaciological and geodetic observations for the test glaciers (1980–2019).

#### 5.1 Performance of MBM on test dataset

The performance of MBM is assessed using the full test dataset from 14 glaciers with in situ mass balance observations. It consists of seasonal and annual point mass balance measurements over the period 1962–2021. MBM shows good performance in predicting both seasonal and annual point mass balances in this test dataset (Figs. 5 and D2, performance on training dataset shown in Fig. D1). Winter mass balance is modelled particularly well, with the lowest RMSE and MAE (0.59 and 0.46 m w.e.; Fig. 5a and b). Summer mass balance is also well captured, and here MBM shows good performance in terms of R<sup>2</sup> (explained variance, 0.72; Fig. 5e). MBM shows somewhat lower performance for annual mass balance in terms of RMSE and MAE (1.00 and 0.77 m w.e.; Fig. 5c), compared to seasonal mass balances, but still with a minimal overall bias (-0.01 m w.e.; Fig. 5d). Overall, the performance of MBM over time is relatively stable, with mean annual and seasonal biases centred around zero (-0.01–+0.04 m.w.e; Fig. 5b, d and f). The second half of the 1970s and 1980s displays some positive bias, but the test dataset contains few measurements in this time period.

Considering point mass balance for glaciers individually (Fig. D2), modelled and observed point mass balances are generally in good agreement, but the performance of MBM varies somewhat between glaciers. It is difficult to compare metrics across glaciers directly due to the different number of point measurements available and varying time periods covered. However, the results do not indicate any particular issues related to climatic region (e.g. continentality in southern Norway or northern versus southern glaciers). Therefore, we are confident that MBM is well suited to capture seasonal and annual point mass balance on glaciers in a wide range of geographical settings in Norway.

#### 5.2 Model comparison on different spatio-temporal scales

We compare predictions from all models (MBM, GloGEM, OGGM and PyGEM) to available glaciological and geodetic mass balance observations for glaciers in the test dataset over the common modelling period 1980–2019. In Sect. 5.2.1 and Sect. 5.2.2, we consider point/elevation-band mass balance and mass balance gradients, respectively, on seasonal and annual time scales. Glacier-wide mass balances are compared in Sect. 5.2.3 on monthly to decadal time scales. We evaluate seasonal and annual predictions using observations from glaciological records (Kjøllmoen et al., 2024), and decadal predictions using

Figure 5. Performance of the Mass Balance Machine on the test dataset of glaciological point mass balance (14 glaciers, 1962–2021), in terms of histograms of errors and temporal biases, respectively, in modelled (a, b) winter, (c, d) summer and (e, f) annual point mass balance. Notations  $b_w$ ,  $b_s$ , and  $b_a$  refer to winter, summer and annual point mass balance, respectively. Points and shaded areas in panels b, d, and f represent the mean and spread of the bias for each year, respectively. Metrics RMSE and MAE in panels a, c and d are in m w.e., and n in panels refers to the number of point measurements.

glaciological and geodetic (Andreassen et al., 2016, 2020; Hugonnet et al., 2021) observations. Over the common modelling period 1980–2019, glaciological observations (point and glacier-wide) are available for 11 of 14 test glaciers with partial temporal coverage. Geodetic observations are available for all 14 glaciers over the period 2000–2019 (Hugonnet et al., 2021), and six sub-periods for four glaciers between 1980–2020 (with an additional six sub-periods and four glaciers back to the 1960s for comparison to MBM only; Andreassen et al., 2016, 2020).

#### 310 5.2.1 Point/elevation-band mass balance

Of all models, MBM shows the best performance with respect to winter and summer point mass balance across all metrics (Fig. 6). Notably, MBM shows very low biases in seasonal mass balance (-0.05 and +0.10 m w.e. for winter and summer, respectively) compared to the glacier evolution models, which show considerable positive or negative biases (+0.26/-0.35/+0.59 m w.e. and -0.27/+0.29/-0.27 m w.e. for OGGM/GloGEM/PyGEM for winter and summer, respectively). Large

**Figure 6.** Modelled point (Mass Balance Machine) or elevation-band (other models) mass balance versus observed point mass balance for the test glaciers for (a) Mass Balance Machine, (b) OGGM, (c) GloGEM and (d) PyGEM, using all available in situ point mass balance observations (n=561/517/516 for annual/winter/summer) over the common model time period 1980–2019. Subscripts a, w and s in legend and metrics refer to annual, winter and summer mass balance, respectively.

positive winter mass balances (> around 2 m w.e.) are particularly well-captured by MBM (Fig. 6a), whereas these are underestimated for GloGEM (Fig. 6c) and overestimated by OGGM and PyGEM (Fig. 6b and d).

GloGEM and OGGM show the overall best performance on annual point mass balance (RMSE of 0.91 and 0.93 m w.e., respectively), but differences between models are relatively small (RMSE of 0.97 and 1.05 m w.e. for MBM and PyGEM, respectively). MBM, GloGEM and OGGM all show low biases (between -0.02 and +0.02 m w.e.), whereas PyGEM displays a relatively large bias in annual point mass balance (+0.36 m w.e.). It should be noted here that MBM is the only model that provides predictions at the point scale, while the other models simulate elevation-band mass balance.

#### 5.2.2 Mass balance gradients





MBM captures mass balance gradients across test glaciers in various climatic settings, from northern Norway (Fig. 7a and b) to the west-east transect in southern Norway (Fig. 7c–f). Mass balance gradients are reproduced particularly well for glaciers in northern and western Norway (Fig. 7a–d). For the most maritime and continental glaciers, modelled mass balance gradients show somewhat larger discrepancy with respect to glaciological observations, for example, positive biases in annual and summer mass balance for high elevations on Hansebreen (Fig. 7e) and in winter mass balance for Hellstugubreen (Fig. 7f). However, it should be noted that for Hellstugubreen, mean elevation-band mass balances are based on only 14 stake measurements over a total of five years, and there are no in situ observations available to investigate model performance at high elevations.

Overall, MBM better captures the relationship between mass balance and elevation than the glacier evolution models over the common period 1980–2019. Again, MBM performs particularly better for seasonal mass balances compared to the glacier evolution models (RMSE/bias of 0.41/+0.1 and 0.60/0.00 m w.e. for MBM for winter and summer, respectively, and best combined scores of 0.56/-0.2 and 0.65/-0.18 m w.e. across other models), while the agreement for annual mass balance gradients is more variable (RMSE/bias of 0.83/+0.08 m w.e. for MBM, and best scores of 0.86/+0.14 m w.e. for other models combined). In general, the glacier evolution models exhibit more linear mass balance gradients than MBM, which in some cases prevents them from capturing the variability in mass balance with elevation (e.g. Svelgjabreen; Fig. 7d).

#### 5.2.3 Glacier-wide mass balance

All models show similar performance in predicting glacier-wide annual mass balance over the common modelling period 1980-2019 (Fig. 8), with MBM performing slightly better in terms of RMSE and R<sup>2</sup> (RMSE of 0.54 m w.e. and R<sup>2</sup> of 0.75; Fig. 8a) and OGGM showing the lowest bias (+0.02 m w.e.; Fig. 8b). There are notable differences in model performance on glacier-wide seasonal mass balance, with MBM outperforming the other models for all metrics and showing particularly better performance for extreme values (high magnitudes). Overall, glacier-wide winter and summer mass balances are overestimated by OGGM and PyGEM (positive and negative bias for winter and summer, respectively; Fig. 8b and d) and underestimated by GloGEM (negative and positive bias for winter and summer, respectively; Fig. 8c).

We evaluate model performances for different regions by aggregating area-weighted glacier-wide mass balance predictions and available glaciological observations over the period 1980–2019 (Fig. D3). Here, we focus on regions North (N) and West (W) (five and six glaciers, respectively; Fig. D3a and c), since an assessment of model performance on regional mass balance is difficult for the most maritime (W-MAR) and continental (E) regions, where glaciological measurements are limited to one and two glaciers in each region, respectively (Fig. D3b and d). All models show relatively good agreement with annual mass balance for region West, where the mass balance rate from glaciological observations over the period 1988–2019 is -0.38 m w.e. a<sup>-1</sup>, compared to -0.35 m w.e. for MBM, -0.31 m w.e. a<sup>-1</sup> for GloGEM, -0.46 m w.e. a<sup>-1</sup> for OGGM and -0.22 m w.e. a<sup>-1</sup> for PyGEM (Fig. D3c). The same is true for region North, with MBM displaying the best correspondence to glaciological observations over the consecutive period 1996–2019 (-0.97 m w.e. a<sup>-1</sup>, versus -0.93 m w.e. a<sup>-1</sup> for MBM,

**Figure 7.** Modelled mean annual (black), winter (blue) and summer (red) mass balance gradients for the Mass Balance Machine (solid lines, with shaded areas showing minimum and maximum predicted elevation-band mass balance between 1980–2019), GloGEM (dashed-dotted lines), OGGM (dashed lines) and PyGEM (dotted lines) for selected test glaciers in different climatic regions over the common period 1980–2019. Circles represent the mean measured mass balance in each 100 m elevation band from available observations over the period. Numbers in subplot titles refer to the last five digits of the RGI 6.0 ID, and abbreviations refer to glacier regions (FIN: Finnmark, BLA: Blåmannsisen, JOB: Jostedalsbreen, FOL: Folgefonna, ALF: Ålfotbreen and JOT: Jotunheimen).

-0.59 m w.e. a<sup>-1</sup> for GloGEM, -0.80 m w.e. a<sup>-1</sup> for OGGM and -0.62 m w.e. a<sup>-1</sup> for PyGEM; Fig. D3a). However, all models show a tendency to underestimate annual mass balance around the 2000s in this region, mostly due to too positive summer mass balance. Considering seasonal mass balance, OGGM and PyGEM show a clear tendency to overestimate magnitudes of winter and summer mass balance in region West, while MBM and GloGEM show good agreement with observations. For

Figure 8. Modelled versus observed glacier-wide seasonal and annual mass balance for the test glaciers for (a) the Mass Balance Machine, (b) OGGM, (c) GloGEM and (d) PyGEM, using all available glaciological observations over the common model time period 1980–2019 (180 mass-balance years on 11 glaciers). Subscripts a, w and s in legend and metrics refer to annual, winter and summer mass balance, respectively.

region North, all models show decent correspondence with winter mass balance. GloGEM shows a tendency to underestimate summer mass balance in the 1990s and 2000s in this region, but better agreement with observations in the 2010s compared to the other models.



We compare monthly glacier-wide mass balance predictions from the four models over the mass balance years 1980–2019 (Fig. 9). Since there are no available mass balance observations at this temporal resolution, we compare monthly glacier-wide predictions for all 32 glaciers in the dataset (15360 predictions per model). For most months, the models show similar mass balance distributions with mostly positive mass balances in Nov–Apr, negative mass balances in Jun–Aug, and both positive and negative mass balances in the transition months May and Sep. This similarity is strongest for Jan–Apr and Oct–Dec. For the summer months, MBM and GloGEM display more moderate mass losses compared to OGGM and PyGEM. MBM's

**Figure 9.** Distributions of glacier-wide monthly mass balance for all 32 training and test glaciers for each month and model over the common time period 1980–2019 (mass-balance years; 15360 monthly predictions per model).

predictions differ somewhat from the other models in the transition months (May and Sep), with a larger number of negative mass balances (see further discussion in Sect. 6.1).

Predicted decadal mass balance rates from MBM show good agreement with decadal rates from glaciological records for most decades (RMSE of 0.26 m w.e.a<sup>-1</sup> and bias of -0.1 m w.e.a<sup>-1</sup> for four glaciers covering 13 decades in total between 1970–2019; Fig. 10a, f, h and m, 1970–1979 not shown for Hellstugubreen, RGI60-08.00449). In general, MBM and the glacier evolution models show similar mass balance rates for many glaciers and decades. However, MBM shows consistently lower mass balance rates for some glaciers, e.g. Langfjordjøkelen, Trollbergdalsbreen and Svartisheibreen (all in northern Norway; Fig. 10a, d and e, respectively), and slightly more positive mass balance rates than the glacier evolution models for others, e.g. Bondhusbrea and Blomstølskardsbreen (Folgefonna; Fig. 10i and l, respectively).

In general, glacier evolution models show a better correspondence with decadal geodetic mass balance rates from satellitederived DEMs (Hugonnet et al., 2021), which is unsurprising given that each test glacier is calibrated using these observations (not independent data). Specifically, MBM overestimates geodetic mass balance for Bondhusbrea, Møsevassbrea and Blomstølskardsbreen (Fig. 10i, k and l, respectively) and underestimates for Langfjordjøkelen and Trollbergdalsbreen (Fig. 10a and d, respectively) when comparing to satellite-borne geodetic mass balance (prediction outside uncertainty bounds for both decades). However, considering geodetic mass balances based on photogrammetry and laser scanning (Andreassen et al., 2016, 2020), MBM shows good correspondence for Langfjordjøkelen (1994–2008; Fig. D4). Overall, all models show decent agreement with geodetic mass balance for Austdalsbreen (region W; 1988–2009) and Hellstugubreen (region E; 1980–1997 and 1997–2010). In addition, MBM shows good agreement for four of six extended periods, including Hellstugubreen (1968–1980), Svartheiisbreen (region N; 1968–2016). Tunsbergdalsbreen (region W: 1964–2013) and Austre Memurubreen (region E: 1966–2009).

#### 6 Discussion





#### 6.1 Performance of MBM and glacier evolution models across spatio-temporal scales

#### 6.1.1 Generalisation through spatio-temporal analogues and downscaling

The ability of MBM to reconstruct glacier mass balance on various spatial and temporal scales demonstrates that ML approaches have the capacity to generalise from mass balance observations at high spatio-temporal resolution and transfer the established relationships to independent (unmonitored) glaciers. We believe that the success of MBM is due to its ability to learn from *spatio-temporal analogues* in training, i.e. similar glacier configurations and/or meteorological conditions across space and time. For example, an abnormally wet winter season on a glacier in northern Norway may be similar to average conditions on a glacier on the west coast of southern Norway, or when a large valley glacier retreats under climate change, its topo-climatic conditions may resemble those of current small, high-altitude glaciers. Using these spatio-temporal analogues, MBM can learn relationships between mass balance and meteorological conditions across diverse climatic settings from relatively sparse data.

The performance of MBM on point mass balance and the apparent importance of the elevation difference feature (see feature importance analysis in Appendix C) suggests that MBM implicitly downscales and bias-corrects relatively coarse meteorological data to the point scale. In addition to the spatio-temporal transfer of mass balance information across glaciers, MBM's apparent downscaling capacity is crucial for generating accurate high-resolution predictions. For instance, MBM's strong performance in reconstructing winter mass balance at the stake level (Fig. 5a and b), together with a high importance of precipitation and elevation difference features in winter months (Fig. C2a–c and k–l), suggests that it is able to downscale precipitation locally. The same is true for temperature in the summer months (Fig. C2e–i). The key to this ability lies in using the elevation difference between the stake and the climate model as a feature (Fig. 3) which enables MBM to effectively map the relationship between climate and elevation.

**Figure 10.** Modelled decadal glacier-wide mass balance rates for each test glacier and model over decadal periods (80s: 1980–1989, 90s: 1990–1999, 00s: 2000–2009, 10s: 2010–2019). Decadal geodetic mass-balance rates with reported error estimates from Hugonnet et al. (2021) are shown as black dots with error bars for the periods 2000–2009 and 2010–2019. Decadal mass balance rates from glaciological records are shown as purple triangles where available. The upper right corner of each panel provide the last five digits of the RGI 6.0 glacier ID and the climatic (N: north, W-MAR: west-maritime, W: west, E: east) and glacier (FIN: Finnmark, SKJ: Skjomen, BLA: Blåmannsisen, SVA: Svartisen, ALF: Ålfotbreen, JOB: Jostedalsbreen, FOL: Folgefonna, JOT: Jotunheimen) regions. Glaciers are ordered from north to south and maritime to continental.

#### 6.1.2 Model evaluation at different spatio-temporal resolutions

The comparison of monthly mass balance (Fig. 9) highlights MBM's ability to make meaningful predictions at a finer temporal resolution than its training data. The largest discrepancy between MBM and the glacier evolution models is observed in May and September, where MBM predicts more negative mass balances. MBM's predictions in these months may be influenced by the definitions of winter and summer seasons, which are based on the median day of the year of the point mass balance measurements in the dataset (5 May and 31 September for winter and annual mass balance measurements, respectively).

However, this definition varies in its alignment with the actual measurement dates, which differs across glaciers and years, potentially leading MBM to compensate with more or less melt or accumulation in the transition months. Improving MBM's monthly predictions could involve using variable season lengths and mass-balance years based on the specific measurement dates in the training data. Nevertheless, when monthly mass balances are aggregated on seasonal scales, MBM shows superior capability in capturing winter and summer mass balance compared to the glacier evolution models across all spatial scales (Figs. 6, 7, 8 and D3).

The ability of MBM to reconstruct winter and summer mass balance on independent glaciers highlights a major advantage compared to the glacier evolution models: MBM does not rely on glacier-specific data and can therefore leverage seasonal mass balance observations to derive relationships that can be transferred to unmonitored glaciers. The glacier evolution models, on the other hand, do not currently use sparse in situ data in their calibration. On annual mass balance the models show similar performance, likely because all models are informed by annual or multi-annual mass balance observations. However, it is important to note that for the glacier evolution models the test glaciers cannot be considered independent in the same respect as for MBM (each test glacier is individually calibrated). Meanwhile, for MBM, the test glaciers serve as independent performance measures across all spatio-temporal scales. Consequently, MBM's performance solely reflects its capacity to generalise to unmonitored glaciers across varying conditions.





Given that the glacier evolution models calibrate parameters for each test glaciers with decadal geodetic mass balance rates from Hugonnet et al. (2021), it is unsurprising that their correspondence to these observations is better than MBM (Fig. 10), which has not employed data from any of these glaciers. However, caution should be taken in interpreting results of this comparison for specific glaciers, since elevation-change rates from Hugonnet et al. (2021) have been found to be substantially lower than those from repeat airborne laser scanning (LiDAR) surveys in Norway (two glaciers, one of which is Austdalsbreen; Fig. 10h; Andreassen et al., 2023). The quality of these geodetic observations, therefore, likely varies between glaciers. For example, for Trollbergdalsbreen (Fig. 10d) MBM shows good performance on point mass balance (Fig. D2d), suggesting that the discrepancy between models may be due to a positive bias in geodetic mass balance from Hugonnet et al. (2021). On the other hand, for Svartisheibreen (Fig. 10e), MBM likely underestimates decadal mass balance rates, as indicated by its relatively strong negative bias on point mass balance for this glacier (Fig. D2e, respectively). Considering Langfjordjøkelen, the performance of MBM seemingly varies over time, since comparisons show both underestimation for some periods and good agreement for others (Figs. 10a, D4 and D2a). However, in decadal comparisons it should be noted that glaciological and geodetic mass balances are not directly comparable, since the latter also include contributions from internal and basal processes that are not accounted for by the models (Zemp et al., 2013; Andreassen et al., 2016, 2023). In addition, when comparing model results to geodetic mass balance from (Andreassen et al., 2016, 2020), predictions are not exactly aligned with survey dates, but aggregated based on the nearest month. For example, for Hansebreen this discrepancy may result in 14 days of more or less melt from the survey date in August 1997, which could be significant considering the high mass turnover of this glacier. This may explain why most models show underestimation and overestimation of the geodetic mass balance rates for 1988–1997 and 1997–2010, respectively (Fig. D4).

## 6.2 MBM design choices and limitations

#### 450 6.2.1 Quality of training data






The quality of ML model predictions is strongly dependent on the quality of the training data, both targets (discussed in Appendix A) and features. While MBM apparently performs well in bias-correcting and downscaling meteorological variables to the elevation of the stakes, it is not always able to perform this downscaling seamlessly for glaciers that span several ERA5-Land grid cells. For example, Tunsbergdalsbreen (RGI60-08.00434), Norway's largest outlet glacier (46.2 km<sup>2</sup> in 2019; Andreassen et al., 2022), is covered by multiple grid cells (Fig. 11d and e), resulting in visible artefacts in the mass balance distribution in some years due to transitions between the uppermost cells (winter and annual mass balance in year 2000; Fig. 11a and b, respectively). These artefacts may occur due to elevation differences not being well represented in the training data, possibly in combination with special meteorological conditions (e.g. decreasing precipitation amounts with elevation, Fig. 11d and f). The relatively coarse resolution of ERA5-Land compared to the extent of most glaciers in our dataset also means that the spatial distribution of mass balance is largely influenced by the higher-resolution topographical information that can resolve smaller-scale variations. Artefacts in the topographical features may therefore influence predictions. For example, MBM predicts high summer melt rates along the eastern border of the tongue of Tunsbergdalsbreen (Fig. 11c). We believe this is due to the combination of steep and south-west facing slopes (Fig. 11g and h). However, these steep, south-west facing slopes are likely topographical artefacts. They result from the surrounding terrain influencing the calculation of these variables from the DEM, specifically a steep, south-west facing wall that borders the glacier tongue. The issues outlined here may be mitigated by extracting meteorological variables from a single ERA5-Land cell closest to the glacier centre. Another option would be to train MBM using higher-resolution meteorological data, which may also elucidate MBM's downscaling capabilities. Regardless of these challenges, our results show that MBM excels in reconstructing local winter mass balance, which indicates implicit downscaling and bias correction of meteorological variables (Figs. 6 and 7). This suggests, in line with other findings (Guidicelli et al., 2023), that ML models are valuable tools to assess spatio-temporal biases in precipitation estimates in mountain regions.

## 6.2.2 Design of test dataset

In addition to the quality of the data used in model training, the predictions and performance evaluation of MBM will be affected by our design of the test dataset and the cross-validation strategy. Although we have attempted to design our test dataset as a reliable measure of our modelling goal, it is not without flaws. For example, the independence between the test and training dataset can be questioned for some glaciers, e.g. for glaciers in the maritime western region of Norway (W-MAR, Fig. 1d). There are only two glaciers from this (small) region in our dataset, of which Ålfotbreen is in the training dataset and Hansebreen (Fig. D2f) is in the test dataset. These glaciers are adjacent, and the spatial correlation between measurements likely extends beyond the ice divide. However, the current configuration is necessary to both train MBM and evaluate its performance in this climatic region. Overall, we did not find any correlation between model performance on the test glaciers and the distance to the nearest training glacier. This is illustrated by the four glaciers in the Folgefonna region (Fig. D2i–I),

**Figure 11.** Distributed (a) annual, (b) winter and (c) summer surface mass balance ( $B_a$ ,  $B_w$  and  $B_s$ , respectively, in m w.e.) for Tunsbergdalsbreen (RGI60-08.00434) in 2000, and selected features: (d) ERA5-Land total precipitation (tp, in mm w.e.) and (e) temperature (t2m, in degC) in February, (f) elevation, (g) slope, (h) aspect and (i) skyview factor (syf, dimensionless).

where the test glacier closest to a training glacier (around 4 km, Fig. D2i) shows worse performance than the glaciers located farther away (up to 12 km, Fig. D2j–l). We encourage future studies using ML approaches to carefully design test datasets using domain knowledge such that performance estimates align with the modelling objectives. However, as illustrated by our example, limitations in the dataset will inevitably require compromises in test dataset design (Roberts et al., 2017).

## 6.2.3 MBM architecture



MBM was designed using the XGBoost architecture due to its excellent performance on tabular datasets (Grinsztajn et al., 2022). However, a known issue with regression tree-based models is that they tend to perform poorly at extrapolation, making them unreliable in accurately capturing extreme conditions beyond their training data (e.g. van der Meer et al., 2025). In the design of the train-test split, we ensure that the training dataset includes years of both high melt and accumulation across a variety of glaciers and climatic settings. As such, MBM is explicitly designed for interpolation rather than extrapolation. Consequently, MBM shows good performance on high magnitudes of winter and summer balances in the test dataset (Figs. 6 and 8). While this approach is appropriate for mass balance reconstruction, making future predictions under potentially unprece-

dented conditions may require a different ML architecture with better extrapolation capabilities, such as a neural network (e.g. Bolibar et al., 2022).

#### 6.3 Future outlook on ML in large-scale mass balance modelling

# 6.3.1 Applications of MBM







The ability of MBM to accurately predict seasonal mass balance on unmonitored glaciers makes it particularly suitable for hydrological applications, especially in glacierised catchments where seasonal observations for glacier-specific calibration of other models are lacking. Another promising application of MBM is to generate distributed mass balance predictions as input to ice flow models. MBM's predictions can account for a changing surface topography simply by updating the topographical features prior to a new prediction. The differentiability of many ML approaches also presents a promising scientific venue in terms of building physics-informed ML glacier models. The MBM architecture could be replaced by a neural network, which would provide differentiability, enabling synchronised calibration and inversion of both glacier ice flow dynamics and surface mass balance. Moreover, the spatial resolution of MBM's predictions is adaptable and determined only by the resolution of the DEM used to extract topographical features. The temporal resolution of MBM is also customizable and can be adapted to produce, for example, weekly or seasonal predictions depending on the desired resolution and computational resources. In our study, we have focused on training MBM for a larger region, leveraging the capacity of ML models to generalise mass balance information to unobserved glaciers. However, MBM can also be tailored to estimate glacier-wide mass balance from glaciological surveys at the individual glacier scale. This may improve glacier-wide glaciological mass balance estimates compared to traditional methods used to interpolate and extrapolate point measurements (e.g. altitude-profile method used in glaciological surveys in Norway; Kjøllmoen et al., 2024).

#### 6.3.2 Reconciling glacier mass balance by learning from diverse datasets

MBM is a scalable model with the potential to be extended to larger regions. We have demonstrated that MBM can capture the spatio-temporal heterogeneity in mass balance for glaciers across climatically diverse regions in Norway. Notably, MBM's current design does not use explicit information about space or time, such that it can essentially be applied at any location and period. However, since the model is trained on meteorological conditions specific to Norway and designed for interpolation within this context, we expect its performance be limited in regions with significantly different climates. Future research into the transferability of ML approaches could clarify the extent of such limitations, for example by testing MBM on glaciers in other regions. For larger, or climatically different regions, we expect MBM to benefit from additional training data. Since in situ observations are not readily available for many regions, the diversity of spatio-temporal analogues and extent of MBM's generalisation capabilities on larger scales remain to be investigated.

On the other hand, the purely data-driven nature of ML approaches makes them uniquely suited to take advantage of the increasing availability of remote sensing-based mass balance datasets (e.g., Belart et al., 2017; Pelto et al., 2019; Hugonnet et al., 2021; Falaschi et al., 2023). This could both alleviate the scarcity of in situ training data and improve model predic-

tions. Additional data could likewise benefit temperature index approaches. However, within the current glacier-specific model calibration frameworks, such data requires regional/global spatial coverage to be readily adopted for calibration in large-scale modelling. In this respect, ML approaches present novel tools for reconciling mass balance estimates from the growing archive of glacier observations, since their flexibility allows for integration of datasets at different spatio-temporal scales in training. We have demonstrated one such approach that leverages observations at different temporal resolutions by training MBM to fit aggregations of monthly mass balance to seasonal and annual targets. Similarly, geodetic mass balance observations (e.g. Hugonnet et al., 2021), could be incorporated into MBM's training by aggregating predictions on glacier-wide and decadal scales. Moreover, model training can account for the reliability of the data by weighing the observations in the loss function according to their confidence levels or using uncertainty-aware learning (Diaconu and Gottschling, 2024). Incorporating diverse and complementary datasets could provide reconciled estimates of glacier mass balance across multiple observational datasets.

Our findings show that ML-based mass balance models have significant potential for unmonitored glaciers due to their flexibility and capacity to generalise from sparse measurements across diverse glaciers, capabilities that complement existing models. As demonstrated here, ML approaches show promise in overcoming some of the limitations of current temperature-index approaches and existing calibration frameworks in large-scale modelling. ML approaches are posed to leverage both existing data as well as growing observational resources from satellite remote sensing to enhance glacier mass balance estimates. In light of our findings, we argue that ML models have significant unexplored potential in glacier mass balance modelling that warrants further investigation.

#### 7 Conclusions






This study presented the Mass Balance Machine (MBM), an ML model designed for regional reconstruction of glacier mass balance up to a point scale and with monthly temporal resolution using topographical and meteorological features. MBM was trained on seasonal and annual glaciological point mass balance measurements from glaciers in various climatic settings in Norway, covering the period 1962–2021. MBM showed good performance in reconstructing glacier mass balance on point to glacier-wide scales for independent glaciers in Norway, demonstrating its ability to generalise spatio-temporal information from sparse data to unmonitored glaciers.

The predictions of MBM were compared to established large-scale glacier evolution models GloGEM, OGGM and PyGEM applied at a regional scale and using current state of the art calibration frameworks. MBM was superior in predicting seasonal mass balance both at point and glacier-wide scales. This success can be attributed to MBM's ability to effectively transfer information from relatively sparse seasonal point mass balance observations to unmonitored glaciers, while current large-scale evolution models do not include these sparse seasonal measurements in model calibration. The glacier evolution models, which rely on multi-year geodetic mass balance for each individual glacier, showed similar performance to MBM on annual and decadal mass balance. The main advantage of MBM is thus that it does not rely on glacier-specific observations and can therefore leverage sparse seasonal data to improve seasonal mass balance predictions across glaciers. The accuracy of

MBM's seasonal predictions suggests that it can improve predictions of seasonal glacier runoff on unmonitored glaciers and thus enhance hydrological modelling in glacierised regions without in situ observations.

The flexibility of ML approaches and their data-driven nature make them uniquely posed to reconcile glacier mass balance from both existing and novel satellite-derived observational datasets at different spatio-temporal scales. We demonstrated that ML models can be adapted to utilise observations at different temporal resolutions by training MBM to fit aggregations of monthly mass balance to seasonal and annual targets. However, there is still significant untapped potential to improve MBM's predictions by incorporating additional data, such as geodetic mass balance observations, in its training.

The ability of ML approaches to learn statistical relationships that are transferable in space and time from high-quality observational datasets provides opportunities to improve mass balance estimates for unmonitored glaciers. Our findings reveal the promise of these approaches in addressing some of the limitations of existing large-scale mass balance models. We advocate for further exploration and development of ML-based mass balance models in order to clarify their contribution to improving glacier mass balance predictions.

Code and data availability. Includes material ©CCME 2024, provided under COPERNICUS by the European Union and ESA, all rights reserved. The Copernicus DEM GLO 90 is available through the Open Global Glacier Model (OGGM) and online at https://doi.org/10.5270/ESA-c5d3d65. Monthly averaged reanalysis data from ERA5-Land (Muñoz Sabater et al., 2021) is available at https://doi.org/10.24381/cds.68d2bb30. The cleaned version of the point mass balance dataset for Norway used in this study (Elvehøy et al., 2025) is available at https://doi.org/10.58059/sjse-6w92. Glacier-wide mass balances from glaciological investigations in Norway can be found at https://glacier.
755 nve.no/Glacier/viewer/CI/en/. This study uses a prototype version of the Mass Balance Machine (MBM) for which the source code and data is available in the GitHub repository https://github.com/khsjursen/ML\_MB\_Norway.git (v.1.0.0 release: https://doi.org/10.5281/zenodo. 15021795). For users interested in the application or further development of MBM, we recommend the official and most recent version at https://github.com/ODINN-SciML/MassBalanceMachine.

Author contributions. KHS and JB conceived and designed the study and methodology. KHS performed the data cleaning and processing, and designed and implemented the machine learning model code with significant contributions from MvdM and JPB and input from JB. KHS prepared the evaluation of the machine learning model and model comparison, with input on the analysis from JB, MvdM. and TD. MH (GloGEM), FM (OGGM), DRR (PyGEM) and BT (PyGEM) performed simulations and provided results from the respective global glacier evolution models. LMA collected and provided the mass balance data and prepared it for publication. KHS prepared the figures and wrote the initial draft of the manuscript with input from JB, MvdM and TD. All authors contributed to the revision and final version of the submitted manuscript.

Competing interests. The authors declare no competing interests.



Acknowledgements. DR was supported by NASA Award 80NSSC24K1530 and BT was supported by the DOI NPS Grant #P22AC02208-02. We would like to thank Bjarne Kjøllmoen and Hallgeir Elvehøy at the Norwegian Water Resources and Energy Directorate (NVE) for collecting, digitising and providing the point mass balance dataset used in this study and for assisting in the quality control of the dataset. We also acknowledge the Western Norway University of Applied Sciences (HVL) for providing computational resources for model training. We thank Brian Kyanjo and an anonymous reviewer for their constructive comments which helped to improve the paper. Finally, we thank Ying Wang, Wahyu Andy Prastyabudi and Jacob Clement Yde for providing constructive comments on an early draft of the manuscript.

#### Appendix A: Data quality and cleaning







The target data for MBM comes from the glaciological records in Norway, many of which have been reanalysed in recent years (Andreassen et al., 2016; Kjøllmoen, 2017, 2022a, b), including comprehensive uncertainty assessments. Uncertainties in stake measurements originate from various sources, such as probing to the previous year's summer surface, displacement and tilting of stakes and errors in snow and firn densities (Zemp et al., 2013). The total contribution of such uncertainties has been quantified 0.08–0.26 m w.e. a<sup>-1</sup> for five of the glaciers in our dataset (Andreassen et al., 2016), and are considered to be of the same order of magnitude for others (Kjøllmoen, 2017). Although errors may occur, glaciological point mass balance records are considered to be of good quality for most glaciers such that we can be confident that MBM is trained on reliable target data.

To ensure the quality of MBM's training data, we performed a thorough cleaning and quality check of the raw point mass balance dataset (4201 entries, NVE database accessed on 12 October 2022) prior to training MBM. This consisted of removing erroneous values and points with missing locations, and verifying stake locations. For each of the point mass balance measurements, the raw data provided an exact and/or approximate stake location (geographical coordinates and elevation). The approximate location is based on the estimated position and elevation of a given stake ID, whereas the exact location is the actual position and elevation of the stake at the time of measurement (e.g. measured using GPS). Position accuracy can vary, in particular for some of the older data.

Seven entries missing both exact and approximate elevations and 23 entries missing both exact and approximate geographical coordinates were removed from the training dataset. For measurements where only the exact location was unavailable, we used the provided approximate locations (329 instances of coordinates and 37 instances of elevation). We estimated the accuracy of the approximate locations based on the 3723/4156 entries where both exact and approximate coordinates/elevations were given. The mean  $\pm$  standard deviation of the absolute difference between the exact and approximate coordinates is  $166 \pm 498$  mm, while for elevations, it is  $24 \pm 71$  m. Thus, for the relatively few measurements missing exact location, we are confident that the approximate coordinates and elevations provide decent estimates of their actual locations. Finally, we converted geographical coordinates from UTM to latitude and longitude format for compatibility with the feature datasets.

For stakes where both summer, winter, and annual mass balance measurements were available for a given year, we corrected for rounding errors where these were present by replacing annual mass balance values by the sum of seasonal values (magnitudes between 0.01–0.03 m w.e.; 255 instances). One measurement with erroneous winter mass balance (unrealistically high; 9.99 m w.e.) was removed. The total number of annual, summer and winter point mass balance observations after cleaning was 3910, 3929 and 3751, respectively, at a total of 4170 stakes.

## **Appendix B: Feature selection**

Our choice of meteorological features is based on two considerations: firstly, to include variables that are relevant for glacier mass balance (accumulation and ablation), and secondly, limiting the number of (confounding) features to ensure explainable results (see Appendix C). Therefore, we selected the main components of the energy balance and other meteorological

variables that are considered the main drivers of mass balance in Norway (e.g. precipitation and temperature for modelling accumulation). We intentionally refrained from using high-level variables that are derived from meteorological conditions, such as snow depth, snow cover and snow melt. The reason behind this is that many meteorological variables in ERA5-Land are highly correlated and mask the underlying meteorological drivers. For example, snow depth and snow melt are highly correlated with total precipitation and 2 m air temperature, respectively. We found that when including snow depth as a feature, total precipitation becomes redundant, although it is an important driver of the evolution of the snow pack. In addition, we did not see a noticeable difference in performance when using a larger set of derived variables or additional meteorological variables (such as wind speed components). Therefore, we opted not to use them both for clarity and simplicity.

We deliberately avoided using explicit temporal and spatial information (e.g. year and geographical coordinates) as features in MBM. Since climate and space are correlated, spatial predictors may mask underlying meteorological drivers (Roberts et al., 2017). In addition, the use of geolocation data may lead to over-fitting to spatial location (Roberts et al., 2017; Meyer et al., 2019), thus deteriorating model predictions outside of the spatial domain on which it is trained. Moreover, we believe that explicit information about time and space (e.g. year and geographical coordinates) should be irrelevant if the model is able to capture mass changes from meteorological features. In contrast to other ML studies using high spatial resolution mass balance data across multiple glaciers (Anilkumar et al., 2023; Guidicelli et al., 2023), we employ a relatively small set of features (e.g. seven compared to the fourteen used by Anilkumar et al., 2023).

#### **Appendix C: Feature importance**





We performed a feature importance analysis on MBM to investigate the importance of different variables on MBM's performance. Since feature importance is complex to interpret and is not adequately represented by any single metric, we based our assessment on different metrics. We calculated weight and gain scores, which represent the total number of times a feature is used in splitting the data in a node and the average improvement in model performance (sum of loss change for each split) in splits where a feature is used, respectively. To complement this analysis, we computed monthly permutation importance for each feature. This involves consecutively permuting (shuffling) the values of each feature, breaking the relationship between the feature and prediction, and assessing the resulting change in model performance. For a given feature and month, the performance change thus represents the effect of feature permutation on the seasonal and annual predictions.

Temperature is overall the most frequently used feature in the trained model (t2m; Fig. C1a). It also scores highest in terms of gain, followed by elevation difference and downward surface solar radiation (elev\_diff and ssrd, respectively; Fig. C1b). The importance of temperature according to the weight and gain scores is not surprising given that both accumulation and melt are strongly influenced by this variable. The combination of lower gain but relatively similar weight of the remaining features may suggest that these are generally used at lower levels of the tree structures, e.g. to distinguish between smaller variability in mass balance for points on the same glacier.

Considering monthly permutation feature importance, elevation difference is an important feature in all months (Fig. C2). In mid-winter (Dec-Mar) total precipitation is the most important feature (tp; Fig. C21 and a-c) and also relatively impor-

tant compared to other meteorological variables in the transition months April, October and November (Fig. C2d, j and k, respectively). This aligns with the fact that solid precipitation is the main contribution to accumulation on glaciers in Norway. In addition, precipitation is likely a key variable in explaining the substantial differences in winter mass balance rates across climatic regions in Norway.



Temperature is the main influence on model performance in the summer season (May–Sep; Fig. C2e–i). In addition, downward solar radiation and forecast albedo are important in May and June (Fig. C2e and f, respectively), which is consistent with the onset of snowmelt and subsequent changes in albedo. Although albedo is coarsely resolved, it may provide larger-scale geographical information about changes in snow cover, which may be why it is also considered somewhat important in mid-winter months. The transition months April and October show less clear importance between meteorological variables (Fig. C2d and j, respectively). This may be because the timing of transitions between seasons varies with latitude, e.g. glaciers in northern Norway may receive a fair amount of snow in April and October.

We caution against placing too much emphasis on the specific details of the feature importance analysis. For example, when assessing permutation importance, correlated features (i.e. skyview factor and slope) may appear to be less important since, even if one feature is permuted, the model can rely on a second correlated feature. However, the main findings of the feature importance analysis presented here are consistent across metrics and physically meaningful with respect to the main meteorological drivers of mass balance on Norwegian glaciers.

**Figure C1.** Feature importance on trained model in terms of (a) weight and (b) gain (t2m: 2 m air temperature, sshf: surface sensible heat flux, slhf: surface latent heat flux, ssrd: downward surface solar radiation, fal: forecast albedo, str: net surface thermal radiation, tp: total precipitation, elev\_diff: elevation difference between climate model and stake, svf: skyview factor). Weight represents the total number of times a feature is used to split the data, summed over all trees. Gain represents the average improvement in model performance (sum of loss change for each split over all trees) in splits which use the given feature. Shaded grey, white and blue background indicates meteorological features, topographical features and elevation difference feature, respectively.

**Figure C2.** Monthly permutation feature importance on the test dataset (t2m: 2 m air temperature, sshf: surface sensible heat flux, slhf: surface latent heat flux, ssrd: downward surface solar radiation, fal: forecast albedo, str: net surface thermal radiation, tp: total precipitation, elev\_diff: elevation difference between climate model and stake, svf: skyview factor). Each feature is permuted on a monthly basis and the resulting change in model performance is computed with respect to the seasonal and annual targets. Shaded grey, white and blue background indicates meteorological features, topographical features and elevation difference feature, respectively.

Figure D1. Performance of the Mass Balance Machine on the training dataset of glaciological point mass balance (18 glaciers, 1962–2021), in terms of histograms of errors and temporal biases, respectively, in modelled (a, b) winter, (c, d) summer and (e, f) annual point mass balance. Notations  $b_w$ ,  $b_s$ , and  $b_a$  refer to winter, summer and annual point mass balance, respectively. Points and shaded areas in panels b, d, and f represent the mean and spread of the bias for each year, respectively. Metrics RMSE and MAE in panels a, c and d are in m w.e., and n in panels refers to the number of point measurements.

**Figure D2.** Performance of the Mass Balance Machine on individual glaciers in the test dataset of glaciological point mass balance (14 glaciers, 1962–2021), in terms of modelled versus measured point mass balance. Subscripts *a*, *w* and *s* in Root Mean Squared Error (RMSE) refer to annual, winter and summer mass balance, respectively. The upper right corner of each panel provide the last five digits of the RGI 6.0 glacier ID and the climatic (N: north, W-MAR: west-maritime, W: west, E: east) and glacier (FIN: Finnmark, SKJ: Skjomen, BLA: Blåmannsisen, SVA: Svartisen, ALF: Ålfotbreen, JOB: Jostedalsbreen, FOL: Folgefonna, JOT: Jotunheimen) regions. Glaciers are ordered from north (FIN) to south (FOL) and maritime (ÅLF) to continental (JOT).

**Figure D3.** Time series of area-weighted glacier-wide annual ( $B_a$ ; grey), winter ( $B_w$ ; blue) and summer ( $B_s$ ; red) mass balance for different models (Mass Balance Machine; solid, OGGM; dashed, GloGEM; dashed-dotted and PyGEM; dotted lines) and regions (a–d). The number of glaciers per region is indicated by n. Area-weighted glacier-wide mass balances from glaciological observations (black solid lines with dots) are shown where observations are available for all glaciers in the region.

Figure D4. Difference between modelled glacier-wide mass balance rates and geodetic mass balance rates for test glaciers and periods with available data (Andreassen et al., 2016, 2020) between 1960–2021 (Mass Balance Machine) and 1980–2019 (all models). Shaded areas show reported uncertainty in geodetic mass balance rates. Modelled mass balance rates are computed between nearest months to geodetic survey dates. Abbreviated names of glaciers (Lan: Langfjordjøkelen, Run: Rundvassbreen, Sva: Svartisheibreen, Han: Hansebreen, Tun: Tunsbergdalsbreen, Aus: Austdalsbreen, Hel: Helstugubreen, Mem: Austre Memurubreen) and subperiod covered (e.g. 66-08 is 1966–2008). Top axis shows the last five digits of the RGI 6.0 glacier ID and the climatic (N: north, W-MAR: west-maritime, W: west, E: east) and glacier (FIN: Finnmark, SKJ: Skjomen, SVA: Svartisen, ALF: Ålfotbreen, JOB: Jostedalsbreen, JOT: Jotunheimen) regions. Glaciers are ordered from north to south and maritime to continental.

# Appendix E: Additional tables

**Table E1.** Summary of model performance metrics on point, mean elevation-band and glacier-wide mass balance using available glacio-logical data for the test glaciers over the common modelling period 1980–2019 (corresponding to Figs. 6 and 8, for point and glacier-wide, respectively). Performance metrics are Root Mean Squared Error (RMSE; m w.e.), bias (m w.e.) and explained variance ( $\mathbb{R}^2$ ; -), and n is the number of data points. Model metrics are highlighted in bold for the best performing models when the performance metric represents an improvement of 5% or more with respect to the next best performing model for RMSE and  $\mathbb{R}^2$  or an absolute reduction of 0.1 m w.e. or more for bias. Mean mass balance in elevation bands is calculated for 100 m bands from available point mass-balance observations, which varies between 6–130 per glacier, giving a total of 55, 55 and 56 mean elevation-band values for annual, summer and winter, respectively.

| Spatial        | Temporal | n   | Metric         | MBM   | GloGEM | OGGM  | PyGEM |
|----------------|----------|-----|----------------|-------|--------|-------|-------|
|                |          |     | RMSE           | 0.55  | 0.65   | 0.82  | 0.87  |
|                | Winter   | 517 | Bias           | -0.05 | -0.35  | 0.26  | 0.59  |
|                |          |     | $\mathbb{R}^2$ | 0.68  | 0.55   | 0.26  | 0.17  |
|                |          |     | RMSE           | 0.70  | 0.79   | 0.89  | 0.93  |
| Point          | Summer   | 516 | Bias           | 0.10  | 0.29   | -0.27 | -0.27 |
|                |          |     | $\mathbb{R}^2$ | 0.68  | 0.60   | 0.49  | 0.44  |
|                |          |     | RMSE           | 0.97  | 0.91   | 0.93  | 1.05  |
|                | Annual   | 516 | Bias           | 0.01  | -0.02  | 0.02  | 0.36  |
|                |          |     | $\mathbb{R}^2$ | 0.64  | 0.69   | 0.68  | 0.59  |
|                |          |     | RMSE           | 0.41  | 0.56   | 0.84  | 0.91  |
|                | Winter   | 180 | Bias           | 0.10  | -0.20  | 0.33  | 0.71  |
|                |          |     | $\mathbb{R}^2$ | 0.80  | 0.62   | 0.15  | -0.01 |
|                | Summer   | 180 | RMSE           | 0.60  | 0.65   | 0.73  | 0.72  |
| Elevation-band |          |     | Bias           | 0.00  | 0.35   | -0.18 | -0.25 |
|                |          |     | $\mathbb{R}^2$ | 0.67  | 0.61   | 0.51  | 0.52  |
|                |          |     | RMSE           | 0.83  | 0.90   | 0.86  | 0.92  |
|                | Annual   | 180 | Bias           | 0.08  | 0.14   | 0.14  | 0.45  |
|                |          |     | $\mathbb{R}^2$ | 0.72  | 0.68   | 0.71  | 0.66  |
|                |          |     | RMSE           | 0.39  | 0.42   | 0.67  | 0.72  |
|                | Winter   | 56  | Bias           | 0.03  | -0.23  | 0.25  | 0.50  |
|                |          |     | $\mathbb{R}^2$ | 0.84  | 0.81   | 0.53  | 0.45  |
|                |          |     | RMSE           | 0.55  | 0.71   | 0.70  | 0.73  |
| Glacier-wide   | Summer   | 55  | Bias           | -0.16 | -0.23  | -0.25 | -0.22 |
|                |          |     | $\mathbb{R}^2$ | 0.73  | 0.55   | 0.56  | 0.53  |
|                |          |     | RMSE           | 0.54  | 0.56   | 0.58  | 0.66  |
|                | Annual   | 55  | Bias           | -0.12 | 0.20   | 0.02  | 0.28  |
|                |          |     | $\mathbb{R}^2$ | 0.75  | 0.73   | 0.71  | 0.62  |

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
