# Peer review of "Machine learning improves seasonal mass balance prediction for unmonitored glaciers"

_EGUsphere, 2025_

## Referee Comment (RC2)

**Review of "egusphere-2025-1206"**

Brian Kyanjo, PhD

May 17, 2025

**Overall Assessment**

The paper "egusphere-2025-1206" (https://doi.org/10.5194/egusphere-2025-1206)
presents the Mass Balance Machine (MBM), a machine learning model for predicting
glacier surface mass balance, trained on a robust dataset of 4,201 point mass balance
measurements from 32 Norwegian glaciers (1962–2021) using the XGBoost algorithm.
This work is a significant advancement in glaciology, offering high-resolution predictions
that outperform traditional models like GloGEM, OGGM, and PyGEM, particularly
for seasonal mass balances. Its potential for applications in climate change research and
water resource management is substantial. However, minor refinements in data resolution,
model transparency, and uncertainty analysis, along with clarifications in Appendix A,
could elevate its impact. Below, I provide a detailed review of the paper's strengths, areas
for improvement, and a focused analysis of Appendix A, including specific corrections and
broader recommendations.

**Strengths of the Paper**

1. **Robust Dataset**: The dataset, sourced from the Norwegian Water Resources and
   Energy Directorate (NVE) database, spans nearly six decades and includes 4,201
   point mass balance measurements across 32 glaciers. The thorough cleaning process,
   detailed in Appendix A, ensures reliability by addressing missing coordinates and
   outliers, resulting in 3,910 annual, 3,929 summer, and 3,751 winter measurements.

2. **Effective Methodology**: The use of XGBoost is well-suited for capturing complex
   relationships between weather, terrain, and glacier mass balance. The independent
   glacier-based train-test split enhances the model's generalizability, making results
   trustworthy.

3. **Superior Performance**: MBM demonstrates lower RMSE and bias compared to
   established models, particularly for seasonal predictions. This is evident in figures
   like Fig. 6 and Table D1, which effectively support the text.

4. **Practical Applications**: The model's high-resolution predictions at point and
   monthly scales are valuable for water resource planning and glacier flow modeling
   in a warming climate.

**Areas for Improvement**

1. **Data Resolution**: The ERA5-Land data (9 km resolution) may be too coarse for smaller glaciers. Exploring higher-resolution datasets or downscaling techniques could improve local accuracy.

2. **Model Transparency**: XGBoost's complexity warrants feature importance analysis or partial dependence plots to clarify key drivers of predictions, enhancing interpretability.

3. **Uncertainty Quantification**: While measurement uncertainties (0.08–0.26 m w.e. $a^{-1}$) are noted, their impact on model outputs is unclear. A sensitivity analysis would strengthen confidence in predictions.

4. **Global Applicability**: Testing MBM in diverse regions like the Alps or Himalayas would broaden its relevance. A discussion of transferability challenges would be valuable.

5. **Future Directions**: The mention of remote sensing data is promising but vague. Specifying datasets (e.g., satellite-derived albedo or surface temperature) would clarify future enhancements.

6. **Presentation Polish**: Minor typos and awkward sentences, particularly in Appendix A, need correction. Additionally, Fig. 10 requires clearer labels for improved readability.

**Detailed Review of Appendix A**

Appendix A details the data quality and cleaning processes for the MBM dataset, critical for establishing its reliability. It describes the handling of 4,201 point mass balance measurements from the NVE database (accessed 12 October 2022), including the removal of erroneous entries and verification of stake locations. Below, I identify specific typos and awkward sentences with approximate line numbers (based on sequential sentence or paragraph counting) and provide broader recommendations to enhance clarity.

**Identified Typos and Awkward Sentences**

1. **Line 570**: "The total contribution of such uncertainties have been quantified 0.08-0.26 m w.e. $a^{-1}$..."

   - *Issue*: Subject-verb agreement error; "have" should be "has" for the singular subject "The total contribution."
   - *Suggestion*: Revise to "The total contribution of such uncertainties has been quantified as 0.08–0.26 m w.e. $a^{-1}$ for five glaciers in our dataset."

2. **Line 575**: "Prior to training MBM, we performed a thorough cleaning and quality check... including removal of erroneous values and points with missing location, and a quality check of stake locations."

   - *Issue*: Redundant use of "quality check."

- *Suggestion*: Streamline to "Prior to training MBM, we performed thorough cleaning and quality checks on the raw point mass balance dataset (4,201 entries, NVE database, accessed 12 October 2022), removing erroneous values, points with missing locations, and verifying stake location accuracy."

3. **Line 578**: "Approximate locations are based on the approximate position and elevation of a given stake ID..."

   - *Issue*: Repetition of "approximate" is awkward.
   - *Suggestion*: Revise to "Approximate locations are derived from the estimated position and elevation of a given stake ID, whereas exact locations use GPS-measured position and elevation at the time of measurement."

4. **Line 581**: "Seven and 23 entries that were missing both exact and approximate elevation or geographical coordinates, respectively, were removed..."

   - *Issue*: Ambiguous phrasing regarding elevation and coordinates.
   - *Suggestion*: Clarify to "Seven entries missing both exact and approximate elevations and 23 entries missing both exact and approximate geographical coordinates were removed from the training dataset."

5. **Line 585**: "The mean ± standard deviation of the absolute difference between the exact and approximate coordinates and elevations is 166 ± 498 m and 24 ± 71 m, respectively."

   - *Issue*: Dense phrasing combines measurements, reducing clarity.
   - *Suggestion*: Split to "The mean ± standard deviation of the absolute difference between exact and approximate coordinates is 166 ± 498 m, while for elevations, it is 24 ± 71 m."

6. **Line 589**: "For stake locations where both summer, winter and annual mass balance measurements were available..."

   - *Issue*: List lacks an Oxford comma for clarity.
   - *Suggestion*: Revise to "For stake locations where summer, winter, and annual mass balance measurements were all available for a given year..."

**Additional Recommendations for Appendix A**

1. **Improve Transitions**: The shift from uncertainties to data cleaning is abrupt. Add a bridging sentence, e.g., "Ensuring dataset quality is crucial for MBM's accuracy, leading to the following cleaning procedures."

2. **Define Technical Terms**: Define "point mass balance" (e.g., "measurements of mass change at specific glacier locations") in a footnote or glossary for accessibility.

3. **Clarify Data Sources**: Specify that NVE is the Norwegian Water Resources and Energy Directorate to aid international readers.

4. **Quantify Cleaning Impact**: State the total entries removed, e.g., "After cleaning, the dataset was reduced from 4,201 to 4,170 stake locations (99.3% retained)."

5. **Explain Coordinate Conversion**: Justify the UTM to latitude/longitude conversion, e.g., "This conversion ensured compatibility with MBM's input requirements."

6. **Justify Erroneous Values**: Explain the removal of the 9.99 m w.e. measurement, e.g., "This value was unrealistically high for typical regional winter mass balances."

7. **Quantify Corrections**: If available, note the number of rounding error corrections, e.g., "In [X] instances, annual mass balances were corrected by summing summer and winter components."

**Conclusion**

The "egusphere-2025-1206" paper is a compelling contribution to glaciology, with MBM offering high-resolution, accurate predictions for glacier mass balance. Its robust dataset, effective methodology, and practical applications make it a valuable tool. Minor revisions, including addressing typos in Appendix A, improving data resolution, and enhancing model transparency, will further strengthen its impact. Appendix A effectively supports the dataset's reliability but can be polished with clearer transitions, defined terms, and quantified impacts. These changes require minimal effort but will significantly enhance the paper's clarity and global relevance.

**Recommendations**

- **Accept with Minor Revisions**.

- **Specific Actions**:

    - Correct the six typos and awkward sentences in Appendix A as suggested.
    - Explore higher-resolution weather data or downscaling for smaller glaciers.
    - Add feature importance or partial dependence plots for model transparency.
    - Conduct a sensitivity analysis to quantify uncertainty impacts.
    - Discuss testing MBM in other regions for global applicability.
    - Specify remote sensing datasets (e.g., albedo, surface temperature) for future work.
    - Add a transitional sentence in Appendix A between uncertainties and cleaning.
    - Define "point mass balance" in a footnote or glossary.
    - Clarify NVE as the Norwegian Water Resources and Energy Directorate.
    - Quantify total entries removed during cleaning (e.g., 4,201 to 4,170).
    - Justify UTM to latitude/longitude conversion and the 9.99 m w.e. removal.
    - Note the number of rounding error corrections, if available.
    - Improve Fig. 10 labels for clarity.
    - Include missing DOIs or URLs in the reference section.

---

## Author Comment (AC1)

*We would like to extend our sincere appreciation for the reviewer's time and dedication in reviewing our manuscript. We thank the reviewer for the positive remarks and for thoughtful and constructive comments. In the following response, reviewer comments are indicated in black and our responses are indicated in blue italic font.*

**Reviewer 1**

The paper by Sjursen et al. introduces the Mass Balance Machine (MBM), a machine learning-based model build on XGBoost, to improve seasonal and annual glacier mass balance predictions across Norway. Using ~4000 in-situ seasonal and annual point measurements from 32 glaciers between 1962 and 2021, the authors demonstrate that the model can generalize well across unmonitored glaciers with diverse climatic settings. MBM outperforms traditional temperature-index glacier evolution models (GloGEM, OGGM, and PyGEM) particularly in predicting seasonal mass balance, reducing RMSE by up to 46% (winter) and 25% (summer). The model performance is robust across multiple spatial and temporal scales, showing strong potential for enhancing hydrological predictions and climate impact assessments in glacierized regions.

I think the MBM is a very promising addition to the traditional glacier evolution models. However, at first instance after reading the manuscript I was questioning to what extent the comparison between MBM and the other models is fair because they are based on different datasets (glaciological versus geodetic) that exhibit very different characteristics. See for instance the recent papers by the GlaMBIE team (2025) and Dussaillant (preprint) who compare and combine different mass balance data sources.  It seems obvious that when comparing to data of type A (which model A is trained with) model A outperforms model B (which is calibrated with data of type B). I was wondering to what extent the authors are comparing models instead of differences between datasets.

Nevertheless, I believe that the fact that the MBM can be trained with the glaciological data and still predict mass balances for unseen glaciers is its key advantage compared to traditional models. I would recommend the authors to emphasize this more and not jump to "straight-forward" conclusions too fast (such as:  *the MBM is better at seasonal predictions*. Yes, it is, but it is also the only model that has seen seasonal data). In addition, I would like to see more support for the selection of features and feature importance.

The manuscript is very well written, and the language is of a high standard. Occasionally, the readability is somewhat reduced by excessive sentence length and accumulation of complex terminology. This particularly applies to the introduction, see an example below. The analysis is well described, and the figures are of high quality.

All in all, I deem the manuscript fit for publication after a major revision. The suggested changes require minimal additional analyses and some textual considerations. Please consider the more detailed list of suggestions below.

*We have carefully considered the comments and made several improvements to the manuscript in accordance with the suggestions. Here, we provide a reply to the general comments and summarize the main changes in the revised manuscript. Below, we provide a point-by-point response to the each of the specific comments.*

*The reviewer highlights that the key advantage of MBM is that it can be trained on glaciological data and therefore can predict mass balance on unmonitored glaciers. This is our view as well. Our intention is to highlight that this capability, which temperature-index approaches are currently lacking at large scales, is exactly what MBM can offer (e.g. in the abstract we highlight the potential of ML to learn relationships that are transferable in space and time, MBMs ability to generalize from sparse data to unseen test glaciers, i.e. unmonitored glaciers). Our conclusion does not suggest that temperature-index approaches cannot perform as well or better than MBM on specific glaciers if they are calibrated using the same data as MBM. Our analysis focuses on the specific application of large-scale (i.e. regional modelling) where glacier evolution models based on temperature-index approaches, that rely on glacier-specific calibration, do not have this option (since most glaciers are unmonitored). In this setting, our results clearly show that MBM outperforms the other models on seasonal predictions, precisely because it does not rely on glacier-specific data and therefore has the ability to leverage the seasonal data. Regarding the fairness of the comparison, we believe that it is fair that the models are compared according to their capabilities on the specific application, i.e. their ability to predict mass balance on unmonitored glaciers in a large-scale context. This includes their advantages and limitations with respect to their ability to leverage existing data, as explained above.*

*We thank the reviewer for pointing out that our reasoning was not clear. We have simplified the language in the abstract and introduction, considering the provided comments to highlight the context of the comparison. In addition, we have amended several formulations throughout the text to clarify our reasoning and address the concerns that have been raised in the general and specific comments. Please consider the suggested excerpts below:*

*Abstract:*

[revised manuscript text omitted]

*In addition to textual changes, we have added a new appendix with analysis of feature importance (Appendix C), as suggested by the reviewer. This includes additional figures (Figs. C1 and C2) using different methods to assess feature importance and discussion of the findings. In addition, we have expanded on our reasoning behind feature selection, as requested by the reviewer. Please see our detailed responses below to each of the specific comments and suggested changes.*

*Again, we would like to express our gratitude to the reviewer for their time and valuable feedback. We believe the suggested changes have significantly improved the manuscript.*

**Abstract**

L9: To assess the advantage MBM's generalization capabilities, --> To assess the advantage of MBM's generalization capabilities,

*Done.*

**1. Introduction**

L39-42: Despite significant efforts ... unmonitored glaciers. This is one such example of a rather long and complex sentence that reduces readability.

*We have amended the formulations at the end of this paragraph and the beginning of the next paragraph to increase readability:*

(around 0.02% of the worlds glaciers; WGMS, 2023). The scarcity of glacier-specific observations has historically posed a ma-
40 jor challenge in  calibrating temperature-index approaches (e.g. Radić and Hock, 2014).  Significant efforts have been made to develop suitable calibration techniques  using limited data (e.g. Radić and Hock, 2011; Huss and Hock, 2015). However, large-scale models still suffer from transferability issues:  they lack efficient frameworks to leverage  sparse in situ observations  for quantifying mass changes on unmonitored glaciers.

45 The  increasing availability of geodetic mass balance observations has recently  alleviated the lack of glacier-specific observations. These observations assess glacier surface elevation changes from time series of satellite-derived digital elevation models (DEMs) over decadal time scales (e.g. Dussaillant et al., 2019; Shean et al., 2020; Hugonnet et al., 2021). Most large-

L67: the fact that point mass balance measurements from glaciological surveys consist of stake measurements hasn't been introduced yet. It is a minor detail, but rewording "each individual stake" to e.g. "each individual mass balance stake" would improve clarity.

*This is introduced already on line 35.*

L72: The term 'generalising' has been used throughout the manuscript but at this point it was unclear to me what you mean by "Generalising from seasonal and annual point mass balance measurements". I now understand that you refer to the generalisation of distributed measurements on different glaciers (spatially), but here the emphasize seems to be on the seasonal versus annual time scales.

*We refer to both spatial and temporal generalisation. We introduce this concept on L56-60, but our formulation was perhaps unclear. We have amended the formulation on L56-60 to improve readability and highlight that we refer to both space and time: "ML models generalise patterns from training data and apply them to make accurate inferences on new, independent data. They can thus learn statistical relationships between mass balance components and topographical and meteorological variables that are transferable across space and time, including to unsurveyed glaciers and years (e.g. Guidicielli et al., 2023)." On L72, the reference to spatial generalization was implicit in our use of "generalizing from .. point" and "high spatio-temporal resolution". However, we have amended the sentence to clarify that we mean generalization both in space and time: "Generalising from seasonal and annual point mass balance measurements offers the potential to provide high temporal resolution distributed mass balance predictions on unmonitored glaciers, ultimately improving runoff predictions from glacierised catchments."*

**2. Mass balance dataset and study area**

L94: To reduce potential confusion regarding the numbers 4170 vs 3910/3929/3751, you may change "4170 stake locations" to "4170 unique stake locations".

*We agree that this is confusing and thank the reviewer for highlighting this since it is important information. The number 4170 refers to unique combinations of stake*

*locations and years. We think the best term to use is "4170 stakes", but specify what we mean by the term by amending the formulation as follows:*

> et al., 2024), to train MBM. The dataset contains measurements at 4170  stakes (unique combinations of locations
> 100    and years) on 32 individual glaciers on the Norwegian mainland (3082/1088  stakes on 22/10 glaciers in south-
> ern/northern Norway; Fig. 1). Each of the 4170 stakes has between one and three readings (annual, summer
> and/or winter), totalling 3910 annual, 3929 summer and 3751 winter point mass balance measurements ( over the
> period 1962–2021; Fig. 2). In all, the 32 glaciers correspond to an area of 343 km$^2$, or ~15% of the total glacierised area in

*We have amended this throughout the manuscript, referring to stakes and point measurements rather than stake locations.*

**3. The Mass Balance Machine (MBM)**

L116: including the design of an independent test dataset.

*Done.*

**3.2 Model targets and features**

*Feature selection, collinearity, and feature importance:* I support your choice of refraining from using climate derivates such as snow depth and snow cover, but I still wonder how you came to this exact choice of climate features. Sensible and latent heat fluxes also depend on other meteorological variables, such as temperature and humidity. Why didn't you for instance use humidity directly? Have you assessed the collinearity within your feature space? I suggest to either include a collinearity assessment in your paper (appendix) or include a statement that this is not relevant or negligible depending on your findings. What made you decide to use net thermal radiation but downward solar radiation? Considering variables like the albedo makes sense based on physical relevance, but how meaningful is the albedo at the 9 km resolution of ERA5-land? In addition, I suspect many point measurements to be inside a single ERA5-land grid cell causing nearest neighbor interpolation to result in non-unique features.

Since you haven't assessed feature importance in your study (or at least not presented in this manuscript), I suggest including more information on your reasoning and considerations in the selection of climate features. To my knowledge, XGBoost returns feature importance of variables, and it would be feasible to include this analysis in the paper.

*The selection of features attempts to find a middle ground between including relevant variables for accumulation and melt, while keeping the number of variables low enough such that the results are explainable in terms of the main components of the energy balance and the meteorological variables that are considered the main drivers of mass balance variability in Norway (temperature and precipitation). Some collinearity between the meteorological variables is inevitable, for example temperature with*

*sensible and latent heat fluxes, skyview factor with slope. In our experience, including additional features did not significantly affect the model performance (possibly due to collinearity). For example, using u- and v-components of the wind speed did not improve the model. We do not expect inclusion of humidity to do so either, especially when already including heat fluxes and precipitation (which we also expect is strongly correlated with this variable).*

*We modified the explanation (first paragraph of Appendix A2) behind the selection of features as follows:*

Our choice of meteorological features is based on  two considerations, including variables that are relevant for glacier mass balance (accumulation and melt), while avoiding an excessive number of (confounding) features to ensure explainable results (see Appendix C). Therefore, we selected the main components of the energy balance and other meteorological variables that are considered the main drivers of mass balance in Norway (e.g. precipitation and temperature for modelling accumulation). We intentionally refrained from using high-level variables that are derived from meteorological conditions, such as snow depth, snow cover and snow melt. The reason behind this is that many meteorological variables in ERA5-Land are highly correlated and mask the underlying meteorological drivers. For example, snow depth and snow melt are highly correlated with total precipitation and 2 m air temperature, respectively. We found that  when including snow depth as a feature,  total precipitation becomes redundant, although it is an important driver of the evolution of the snow pack.  In addition, we did not see a noticeable difference in performance when using a larger set of derived variables  or additional meteorological variables (such as wind speed components). Therefore, we opted not to use them both for clarity and simplicity.

*The addition of a feature importance analysis is an excellent addition and we thank the reviewer for suggesting this. We added a new appendix (Appendix C: Feature importance) with a discussion of feature importance based on different methods, including two new figures showing overall feature importance in terms of weight and gain on the trained model, and monthly permutation feature importance on the test dataset. The analysis provides additional insights into the importance of different monthly features in seasonal and annual predictions, and we believe that many of the findings support the current assessment of MBM's capabilities. We suggest the following additional Appendix C2, including Figures C1 and C2:*

**Appendix C: Feature importance**

[revised manuscript text omitted]

*It is true that many point measurements are within a single ERA5-Land cell, such that features are in many cases non-unique on the same glacier and month. This is where the elevation difference feature and topographical features become important. These are unique to the given stake location and helps the model to reconstruct mass balance in a sub-climate model resolution. With regards to the albedo, we agree that a resolution of 9km is too coarse to resolve variations in albedo on the glacier. However, we included albedo since it may provide information about snow cover conditions on larger geographical scales (i.e. fresh or wet snow). In our opinion, this is supported by the feature importance analysis (importance of albedo in winter months and transition seasons). When including the albedo as a feature we believe it is more physically accurate to combine this with downward solar radiation instead of net solar radiation.*

L163-166: It is unclear to me how your model learns to predict monthly variability in mass balance. How can you be sure that the monthly predictions make sense? Since there is never any overlap in your seasonal mass balance measurements, couldn't *equifinality* still play a role?

*The model learns monthly variability by considering the monthly meteorological data to make predicting monthly mass balance predictions, which are aggregated and evaluated on the seasonal and annual time scales (Fig. 4). In a sense, this is similar to how temperature-index models make monthly predictions and are calibrated: they are provided monthly meteorological data and predictions are* **aggregated** *to the temporal resolution of the observations before they are compared. We agree that there is certainly a chance that equifinality plays some role in MBM, in terms of monthly predictions compensating each other on the seasonal time scale. For example, we would expect that if flexible dates were used to define summer and winter seasons (instead of the 1 May and 1 October limits that were used in the study), the distributions of monthly mass balances would shift somewhat, but still produce the similar seasonal results. The advantage of MBM is that it can utilize the seasonal data to reduce equifinality (i.e. compensating effects of melt and accumulation is reduced compared to using annal or multi-year mass balance). Unfortunately, we do not have data at the monthly time scale to validate predictions of any of the models. Lacking such data, the intention behind our monthly comparison is to benchmark monthly predictions across models (L239-241). In our opinion, the similarity between the monthly distributions in Fig. 9 is strong evidence that they do make sense. In addition, we believe the newly added feature importance analysis in Appendix C provides support for the physical basis of the monthly predictions.*

**3.3 Model training and testing**

While in L191-192 you state that "The performance evaluation of MBM on the test dataset thus reflects the model's ability to predict mass balance on glaciers without mass balance observations", you did make sure that the distribution of both targets and

features in the train versus test dataset are similar. Is this fair? It is no surprise to me that your model can predict the mass balance on unseen glaciers 'as long as they exist in the same distribution…' In reality, you cannot be sure that the target of an unseen glacier fits into the distribution of targets in your training dataset, you could only know this for the features.

*It is true that we cannot know that the distribution of targets in the dataset reflects the distribution of mass balance for all Norwegian glaciers over the time period. As emphasized in the manuscript, the goal is to design MBM such that it can predict mass balance on all glaciers in Norway. The underlying assumption here is that the dataset (features and targets) reflect the true distribution of glaciers in Norway (data is identically distributed). As we have argued in the manuscript, the spatiotemporal coverage of the dataset used in the study provide a solid representation of the glacier population in Norway. Ensuring that the distribution of the training and test datasets are similar is not unfair, but reflects that each of these datasets are assumed to be drawn from this (unknown) true distribution. This is a common assumption in machine learning (together with data independence discussed in the manuscript it forms the independent and identically distributed (i.i.d.) assumption) and is why the test performance can be seen as reflecting the model's ability to generalize.*

L215-216: If I understood correctly, the location of the stake measurements is not constant throughout the years (since you mention 4170 stake locations, but only up to 200 annual mass balance measurements per year). I assume that this reflects the displacement of a stake due to glacier flow? This usually being only a small displacement, I do not expect the topographic features to vary greatly through time. Therefore, by splitting the data in the 5-fold cross validation only based on time, I expect this to reduce the apparent importance of the topographic features. Have you considered this? Would this affect the hyperparameter tuning?

Our use of the term "stake locations" was confusing and we have specified it to combinations of stake locations and years (please see comment on L94). Stakes are usually redrilled in approximately the same location and the new location is measured using GPS. We would expect a small displacement of the stake over the year, but this should be well within the resolution of the DEM we use to extract the topographical features (90 m) such that these do not vary greatly for a stake in the same approximate location. We do not believe that splitting the data for cross-validation based on time has a significant impact on the importance of the topographical features. The topographical features are intended to determine the mass balance on a sub-climate model scale and their importance will depend on their effect on the model in combination with the meteorological features. We tested several ways of splitting the data for cross validation (including splits on years and random splitting) and did not find that this had a substantial impact on the best hyperparameter combination. In addition, the final

model (using best hyperparameters) is retrained on the full training dataset, such that the final feature importance is determined in this training.

L231: how is the R2 metric computed? Why are you comparing four different metrics but not the MSE that was used in cross-validation?

*The $R^2$ metric is also called the coefficient of determination and is computed as:*

$$R^2 = 1 - \frac{\sum_i (y_i - \hat{y}_i)^2}{\sum_i (y_i - \bar{y})^2},$$

*Where $y_i$ are observed values, $\hat{y}_i$ are predicted values and $\bar{y}$ is the mean of observations. The $R^2$ is a measure of the portion of the variance of the data that is explained by the model.*

*Training the model to minimize the MSE is essentially the same as minimizing the RMSE (the RMSE is just the square-root of the MSE, the squared difference observations and predictions are minimized in both cases). We provide RMSE instead of MSE as one of the four metrics used to evaluate and compare the models because is easier to interpret since it has the same units as the predictions (in this case, m w.e.). Overall, the four metrics provide complementary information about the fit of the models: bias, explained variance and errors (with RMSE giving more weight to outlier errors than MAE).*

**4. Mass balance model comparison**

L247: Unclear what "these glaciers" refers to: the whole test dataset, 11 of the 14 glaciers or the three glaciers referred to in brackets.

*This is a good point and we have now specified that we are referring to the test glaciers.*

L252-L253: is the spatial resolution in table 2 the width/height of the elevation bands? I suggest referring to this more explicitly. From what I understand, GloGEM and OGGM use a fixed vertical spacing (elevation) while PyGEM uses a horizontal spacing (distance).

*We agree that this is important to clarify and have added footnotes to Table 2 explaining the difference.*

I am wondering to what extent the resolution of these elevation bands can explain the differences in performance of the different models. How does the point elevation at the mass balance stakes compare to eg the average elevation of the model elevation bands? For instance, if for whatever reason or by coincidence the stakes are typically located at the higher end of the elevation bands, this would explain the model underestimating the mass balance.

*We agree that the different spatial resolution of the models may affect the results somewhat, but we do not believe that this is the main explanation of the difference*

*between the models. We would mainly expect this to influence the point mass balance comparison (Fig. 6). However, since the overall differences in model performances in the glacier-wide comparison (Fig. 7) are similar to those of the point mass balance comparison, we do not expect the difference in vertical resolution to be a major contributor to these differences. In addition, the vertical resolution of GloGEM and OGGM are relatively high, such that the elevation differences would amount to +/- 5 to 15 meters (see for example first histogram below of vertical distance to nearest bin centers for test points using GloGEM bin centers), which we consider to be too small to have any major influence on the mass balance. Since PyGEM uses horizontal distances along a flowline, the vertical resolution will be higher in flat areas and coarser in steep areas on the glacier, such that we would expect that these effects may be more influential in steeper parts of the glacier. However, since point mass balance measurements are mainly performed on flatter areas due to accessibility, this likely does not have a major impact on the point mass balance comparison here. We also checked the vertical distance between point measurements at different elevations with respect to 100m bins used in Fig. 7 (second histogram below and example plot for Langfjordjøkelen). There is a slightly higher frequency of points at +40 m elevation, but the bulk of the distribution is around 0 to -20m elevation. Our analysis did not show any strong evidence of these differences influencing the comparison. For example, at 850 m elevation, point measurements are generally at 30-40m higher elevation than bin centers, but nevertheless match quite well with all models (Fig. 7a).*

[Figure]

Table 2: Include Tcorr in the list of parameters for GloGEM and include the annotation [e] there. In caption: [e] only included if no match is found with other parameters within predefined bounds.

*Done.*

**5.2 Model comparison on different spatio-temporal scales**

L297-300: This sentence is confusing and the word 'glacier-wide' is often repeated. Glacier-wide mass balances are compared on different time scales. You evaluate glacier-wide predictions using seasonal and annual glacier-wide observations from glaciological records AND you evaluate decadal predictions using glacier-wide glaciological and geodetic observations. Reword to:

"Glacier-wide mass balances are compared in Sect. 5.2.3 on monthly to decadal time scales. We evaluate seasonal and annual predictions using observations from glaciological records (Kjollmoen et al., 2024), and decadal predictions using glaciological and geodetic (Andreassen et al., 2016, 2020; Hugonnet et al., 2021) observations."

*Done. We agree that this improves clarity and thank the reviewer for the suggestion.*

Figure 6: measured --> observed point mass balance

*Changed wording in the caption from "measured" to "observed" to align with the axis labels.*

L330-331: In contrast to the glacier evolution models who exhibit too linear gradients, it seems that the MBM can predict unlikely variability in the gradients. See for instance the knickpoints at higher elevation in Figure 7a and c. These do not seem to correspond to the observations (there is no data point at this elevation). Can you explain the occurrence of such knickpoints?

*Here, we assume that the reviewer refers to the knickpoints on predicted annual and winter mass balance at the highest elevations in Fig. 7a and on predicted annual and summer mass balance in Fig. 7c. We expect these to be artefacts of the model due to coarsely resolved climate data and the lack of measurements (training data) at higher elevations on many glaciers (similar to what can be seen on Tunsbergdalsbreen in Fig. 11). Such artefacts may be mitigated by higher resolution climate data or extracting climate data from a single ERA5-Land cell (instead of using all cells that cover the glacier), such as described in Section 6.2.1. However, not all such knickpoints are unlikely. For example, for annual and winter mass balance at the highest elevation in Fig. 7c, point measurements indicate reductions in annual and winter mass balance that the models are not able to predict and that are perhaps the result of redistribution of snow by wind at higher elevations.*

Figure 7: the almost vertical lines in 7f demonstrate the equifinality issue with the glacier evolution models being calibrated with glacier-wide 20-year average geodetic data and no way of knowing whether there is a shallow or steep mass balance gradient.

*We agree with this observation and believe that this illustrates the advantage of MBM being able to use sparse point mass balance data on different glaciers, compared to temperature-index approaches, which are dependent on calibration to observational data at the scale of individual glaciers (for which 20-year average geodetic data is currently the option). Please see our reply to comment on L414 and reply to general comments.*

L371-372: This is a fair point, but the opposite is also true. The predictions by MBM correspond better to the glaciological observations because they are trained using this

data. Even though you test the model on unseen glaciers, you still train the model using data with similar variability, while the glacier evolution models are calibrated with a 20-year average and will never learn the interannual variability. This could be emphasized more.

*The specific formulation on L371-372 refers to the rigorousness of the comparison of model performances on this specific dataset, not the underlying explanations for why performance of the models differ on different spatiotemporal scales. The performance comparison on the 20-year geodetic data  is not very rigorous for the glacier evolution models since the same data was used to calibrate these models (could be viewed as showing MBM's performance on its training data), as opposed to being an independent dataset for MBM. We included a more rigorous comparison at decadal time scales in Fig. C4, where the dataset is independent in space and time for MBM and independent in time for the glacier evolution models. Thus, we do not agree that the opposite is true with respect to MBM since it is always compared on independent data.  However, we realize that this was not clear and thank the reviewer for pointing that out. We have reformulated the sentence on L371-372 and sentences in  Section 6.1.2 that refer to the same, please see excerpts below.*

*However, we agree that MBM is better at reconstructing seasonal and annual mass balance because it has been trained using this data, while the glacier evolution models have not (and cannot use this effectively in large-scale modelling). Please see our reply to comment on L414 and reply to general comments.*

> In general, glacier evolution models show a better correspondence with decadal geodetic mass balance rates from satellite-derived DEMs (Hugonnet et al., 2021), which is unsurprising given that each test glacier is calibrated using these observations (not independent data). Specifically, MBM overestimates geodetic mass

> The ability of MBM to reconstruct winter and summer mass balance on independent glaciers highlights  a major advantage compared to the glacier evolution models: MBM does not rely on glacier-specific data and can therefore leverage seasonal mass balance observations to derive relationships that can be transferred to unmonitored glaciers.
> 430  The glacier evolution models, on the other hand, do not currently use sparse in situ data in their calibration. On annual mass balance  the models show similar performance.  likely because all models are informed by annual or multi-annual mass balance observations. However, it is important to note that for the glacier evolution models  the test glaciers cannot be considered independent in the same respect as for MBM (each test glacier is individually calibrated). Meanwhile, for MBM, the
> 435  test glaciers serve as independent performance measures across all spatio-temporal scales. Consequently, MBM's performance solely reflects its capacity to generalise to unmonitored glaciers across varying conditions.
>    Given  that the glacier evolution models calibrate parameters for each test glaciers with decadal geodetic mass balance rates from Hugonnet et al. (2021), it is unsurprising that their correspondence to these observations is better than MBM (Fig. 10), which has not employed data from any of these glaciers. However,
> 440  caution should be taken in interpreting results of this comparison for specific glaciers, since elevation-change rates from Hugonnet et al. (2021) have been found to be substantially lower than those from repeat airborne laser scanning (LiDAR) surveys in Norway (two glaciers, one of which is Austdalsbreen; Fig. 10h; Andreassen et al., 2023). The quality of these geodetic observations, therefore, likely varies between glaciers. For example, for Trollbergdalsbreen (Fig. 10d) MBM shows good performance on point mass balance (Fig. D2d), suggesting that the discrepancy between models
> 445  may be due to a positive bias in geodetic mass balance from Hugonnet et al. (2021). On the other hand, for Svartisheibreen

L373-375: Please consider the uncertainties of the geodetic data. I suspect the over- or underestimation of the models to still be within the 95% confidence bound of the geodetic data.

*We are unsure if this comment refers to inclusion of uncertainty estimates in the Figs. 10 and C4, or if our consideration of over- or underestimation with respect to the uncertainty bounds in Fig. 10 is vague. Since uncertainty in the geodetic mass balance from Hugonnet et al. (2021) and geodetic mass balance from NVE (reported 1-sigma uncertainties for both datasets) is already shown in Figs. 10 and C4, we assume the comment refers to the latter. The specific examples of over- or underestimation mentioned here are cases where MBMs predictions are outside the 1-sigma uncertainties for both decades in Fig. 10. We have now specified this in the text by amending the formulation as follows:*

Specifically, MBM overestimates geodetic mass balance for Bondhusbrea, Møsevassbrea and Blomstølskardsbreen (Fig. 10i, k and l, respectively) and underestimates for Langfjordjøkelen and Trollbergdalsbreen (Fig. 10a and d, respectively) when com-
385 paring to satellite-borne geodetic mass balance (prediction outside uncertainty bounds for both decades). However, considering

**Discussion**

L394: How can you be sure that MBM effectively downscales the meteorological data instead of relying on the high-resolution topographic features? Is there any way to support this statement? A feature importance analysis may have provided more insights in this. Alternatively, although this is most probably not within the scope of this manuscript, one could have compared the performance of MBM with coarse meteorological data + elevation difference to already downscaled meteorological data. Or you could have explicitly learned the MBM to downscale climate data using some high-resolution climate variable as additional target. It may have been that elevation difference "appears" to be important because it is one of the few variables that are actually unique for each stake location. Without any support, I question whether you can make the statement that MBM effectively downscales. Especially with regards to Figure 11.

*The newly added feature importance analysis (Appendix C) shows that elevation difference is an important feature, which we would expect since it is the main variable that relates the climate data to the resolution of the point measurements. If the downscaling was done manually, the elevation difference between the climate model and point location would also be very important. Hence, it is It natural that this variable is frequently used in MBM and considered important. Monthly temperature and precipitation are the most important variables in the winter and summer months, respectively (Fig. C2), in addition to elevation difference. We think this indicates that it is implicitly used to downscale these variables to the higher-resolution grid. However, we*

*agree that the statement we made is perhaps too strong, so we moderated our statements in Section 6.1.1. as follows:*

> 400  The performance of MBM on point mass balance and the apparent importance of the elevation difference feature (see feature importance analysis in Appendix C) suggests that MBM implicitly downscales and bias-corrects relatively coarse meteorological data to the point scale. In addition to the spatio-temporal transfer of mass balance information across glaciers, MBM's apparent downscaling capacity is crucial for generating accurate high-resolution predictions. For instance,  MBM's strong performance in reconstructing winter
> 405 mass balance at the stake level (Fig. 5a and b), together with a high importance of precipitation and elevation difference features in winter months (Fig. C2a–c and k–l), suggests that it is able to downscale precipitation locally. The  same is true for temperature in the summer months (Fig. C2e–i). The key to this ability lies in using the elevation difference between the stake and the climate model as a feature (Fig. 3) which enables MBM to effectively map the relationship between climate and elevation.

*We agree that an interesting avenue for future development of MBM is to compare its performance for different resolution climate data. Similar work is already ongoing in applications to other regions in Europe and we expect MBM to benefit from using higher-resolution meteorological data. We suggest this in Section 6.2.1, but have now specified that higher-resolution meteorological data can clarify the downscaling-capabilities of MBM:*

> 465 variables from the DEM, specifically a steep, south-west facing wall that borders the glacier tongue. The issues outlined here may be mitigated by extracting meteorological variables from a single ERA5-Land cell closest to the glacier centre. Another option would be to train MBM using higher-resolution meteorological data, which may also elucidate MBM's downscaling capabilities. Regardless of these challenges,  our results show that MBM excels in reconstructing local winter mass balance, which indicates implicit downscaling and bias correction of meteorological variables
> 470  (Figs. 6 and 7). This suggests, in line with other findings (Guidicelli et al., 2023), that ML models are valuable tools to assess spatio-temporal biases in precipitation estimates in mountain regions.

L414: I think it is important to distinguish between and not confuse two different assets of your model: 1) it can predict mass balances for unmonitored glaciers while the glacier evolution models need calibration data for every single glacier, and 2) it is trained with seasonal and annual data while the glacier evolution models were only provided on single 20-year average value. I think the first point is the big advantage of the MBM and this should be highlighted more, while the second point is an artifact of the first. Because regular models need data for every glacier it cannot be calibrated with the higher temporal resolution data because this is only available for a limited number of glaciers.

*We do not completely agree with this reasoning. The reason that MBM can predict seasonal mass balance for unmonitored glaciers is specifically because it can be trained on seasonal data. For the type of application/ regional modelling of a large set of glaciers with sparse in situ measurements: 1) machine learning models can learn relationships from data and do not require data specific to a given glacier to learn relationships between climate and mass balance on that glacier, while the glacier evolution models need calibration data specific to the given glacier to learn these*

*relationships. 2) Machine learning models can therefore be trained using sparse, in situ data (e.g. point mass balance in this case), while the glacier evolution models rely on datasets available for all glaciers (it is true that for the glaciers with glaciological data, the glacier evolution model could be calibrated to these specific glaciers, but that option does not exist for the vast majority of "unmonitored" glaciers). 3) Since MBM is able to use the seasonal data it can improve seasonal predictions compared to the glacier evolution models, which cannot use this data effectively.*

*To clarify our reasoning we have made several updates to the text throughout the manuscript (please see our reply to the general comments), in addition to the following changes in the section related to this specific comment:*

> The ability of MBM to reconstruct winter and summer mass balance on independent glaciers highlights  a major advantage compared to the glacier evolution models: MBM can leverage seasonal mass balance observations to derive relationships that can be transferred to unmonitored glaciers. On annual mass balance  the models show similar performance. , likely because all models are informed by annual or multi-annual mass balance
> 430 observations. However, it is important to note that for the glacier evolution models  the test glaciers cannot be considered independent in the same respect as for MBM (each test glacier is individually calibrated). Meanwhile, for MBM, the test glaciers serve as independent performance measures  across all spatio-temporal scales. Consequently, MBM's performance solely reflects its capacity to generalise to unmonitored glaciers across varying conditions.

L451-453: It is unclear what you mean. How does the steep terrain influencing the tongue affect the more negative mass balance for steep and south-facing slopes?

*We thank the reviewer for pointing out that this explanation was not clear. The steep terrain around the glacier tongue influences the calculation of slope and aspect from the DEM, such that the border of the tongue is seemingly steep and southwest-facing (while in reality it is flatter and more southeast-facing (like the remainder of the tongue). These artefacts in the topographical features influence MBM's predictions (which are likely too negative here). We have changed the formulation in order to clarify this explanation:*

> can resolve smaller-scale variations. Artefacts in the topographical  features may therefore influence predictions. For example, MBM predicts high summer melt rates along the
> 460  eastern border of the tongue of Tunsbergdalsbreen (Fig. 11c).  We believe this is due to the combination of steep and  south-west facing slopes (Fig. 11g and h). However, these  steep, south-west facing slopes are likely topographical artefacts. They result from the  surrounding terrain influencing the calculation of these variables from the DEM, specifically a steep, south-west facing wall that borders the glacier tongue. The issues outlined

L463: In my opinion, it is not necessary a bad thing to assess the capability of your model in 'extrapolating' to adjacent glaciers. It would be interesting to include a little more of your findings regarding the ability to extrapolate in relation to the distance away from the nearest 'seen' glacier.

*We thank the reviewer for suggesting this interesting analysis that we had not performed. We compared the performance of MBM on the test glaciers to the distance between*

*each test glacier and the nearest training glaciers. The first figure below shows all the test glaciers, while the second figure excludes the two glaciers in northern Norway that have large distances to the nearest glacier. Interestingly, the analysis does not show any correlation between the distance to the nearest training glacier and test glacier performance (which is also what we aim for since spatially correlated errors would suggest that our test glaciers are not independent). This is for example also illustrated by comparing Figs. D2 i-l (test glaciers in the Folgefonna region). Of these glaciers, the glacier in Fig. D2i shows the worst performance, but is the closest to a training glacier (around 4 km), while the other glaciers (Figs. Dj-l) show better performance but are up to 12 km away from the nearest training glacier). We think this analysis suggests that model performance on the test glaciers is more closely related to how well the trained model captures the relationship between mass balance and meteorological and topographical features in different regions of the feature space.*

*Based on this finding, we included the following in Section 6.2.2.:*

485   likely extends beyond the ice divide. However, the current configuration is necessary to both train MBM and evaluate its performance in this climatic region. Overall, we did not find any correlation between model performance on the test glaciers and the distance to the nearest training glacier. This is illustrated by the four glaciers in the Folgefonna region (Fig. D2i–l), where the test glacier closest to a training glacier (around 4 km, Fig. D2i) shows worse performance than the glaciers located farther away (up to 12 km, Fig. D2j–l). We encourage future studies using ML approaches to carefully design test datasets

---

## Author Comment (AC2)

*We thank the reviewer, Brian Kyanjo, for the effort in assessing our manuscript and for the positive feedback and suggestions for improvement. We have considered the feedback carefully and made several changes to the manuscript in accordance with the comments provided. The main improvement is the addition of a feature importance analysis in Appendix C (including two new figures, Fig. C1 and C2), where we consider and discuss feature importance based on different metrics. In addition, we have made several updates to the text in relation to the specific comments. Please see our detailed responses below to each of the specific comments. In the following response, reviewer comments are indicated in black and our responses are indicated in blue italic font.*

**Reviewer 2:**

**Overall Assessment**

The paper "egusphere-2025-1206" (https://doi.org/10.5194/egusphere-2025-1206) presents the Mass Balance Machine (MBM), a machine learning model for predicting glacier surface mass balance, trained on a robust dataset of 4,201 point mass balance measurements from 32 Norwegian glaciers (1962–2021) using the XGBoost algorithm. This work is a significant advancement in glaciology, offering high-resolution predictions that outperform traditional models like GloGEM, OGGM, and PyGEM, particularly for seasonal mass balances. Its potential for applications in climate change research and water resource management is substantial. However, minor refinements in data resolution, model transparency, and uncertainty analysis, along with clarifications in Appendix A, could elevate its impact. Below, I provide a detailed review of the paper's strengths, areas for improvement, and a focused analysis of Appendix A, including specific corrections and broader recommendations.

**Strengths of the Paper**

1. Robust Dataset: The dataset, sourced from the Norwegian Water Resources and Energy Directorate (NVE) database, spans nearly six decades and includes 4,201 point mass balance measurements across 32 glaciers. The thorough cleaning process, detailed in Appendix A, ensures reliability by addressing missing coordinates and outliers, resulting in 3,910 annual, 3,929 summer, and 3,751 winter measurements.

2. Effective Methodology: The use of XGBoost is well-suited for capturing complex relationships between weather, terrain, and glacier mass balance. The independent glacier-based train-test split enhances the model's generalizability, making results trustworthy.

3. Superior Performance: MBM demonstrates lower RMSE and bias compared to established models, particularly for seasonal predictions. This is evident in figures like Fig. 6 and Table D1, which effectively support the text.

4. Practical Applications: The model's high-resolution predictions at point and monthly scales are valuable for water resource planning and glacier flow modeling in a warming climate.

**Areas for Improvement**

1. Data Resolution: The ERA5-Land data (9 km resolution) may be too coarse for smaller glaciers. Exploring higher-resolution datasets or downscaling techniques could improve local accuracy.

*We agree that the resolution of ERA5-Land is coarse compared to the size of the glaciers in the dataset. Our intention with using a globally available dataset is that the model can easily be adapted to other regions. In addition, ERA5-Land is relatively high resolution compared to other globally available datasets. Some higher-resolution climate datasets exist for Norway (e.g. the NORA datasets), but these do not cover the entire time period of the mass balance measurements. However, in terms of resolution, we believe that MBM can, at least to some degree, implicitly downscale the meteorological data to the elevation of the point measurements by using the elevation difference feature to distinguish between points within the same ERA5-Land cell. This is evidenced by MBM's performance on such data (Figs. 5 and 6) and supported by the newly added feature importance analysis. We have elaborated on this in Section 6.1.1:*

> 400     The performance of MBM on point mass balance and the apparent importance of the elevation difference feature (see feature importance analysis in Appendix C) suggests that MBM implicitly downscales and bias-corrects relatively coarse meteorological data to the point scale. In addition to the spatio-temporal transfer of mass balance information across glaciers, MBM's apparent downscaling capacity is crucial for generating accurate high-resolution predictions. For instance,  MBM's strong performance in reconstructing winter
> 405    mass balance at the stake level (Fig. 5a and b), together with a high importance of precipitation and elevation difference features in winter months (Fig. C2a–c and k–l), suggests that it is able to downscale precipitation locally. The  same is true for temperature in the summer months (Fig. C2e–i). The key to this ability lies in using the elevation difference between the stake and the climate model as a feature (Fig. 3) which enables MBM to effectively map the relationship between climate and elevation.

*It would be interesting to compare the performance of the model on other climate datasets and different resolutions, and we expect that the performance of both MBM and the other models to increase with increasing climate data resolution. We mentioned the use of higher-resolution meteorological data already in Section 6.2.1 as an option that may improve MBM's predictions (and limit reliance on high-resolution topographical features). We consider this to be out of the scope of the current study, but have elaborated some more in Section 6.2.1:*

can resolve smaller-scale variations. Artefacts in the topographical  features may therefore influence predictions. For example, MBM predicts high summer melt rates along the  border of the tongue of Tunsbergdalsbreen (Fig. 11c).  We believe this is due to the combination of steep and  south-west facing slopes (Fig. 11g and h). However, these  steep, south-west facing slopes are likely topographical artefacts. They result from the  terrain  surrounding terrain influencing the calculation of these variables from the DEM, specifically a steep, south-west facing wall that borders the glacier tongue. The issues outlined here may be mitigated by extracting meteorological variables from a single ERA5-Land cell closest to the glacier centre. Another option would be to train MBM using higher-resolution meteorological data, which may also elucidate MBM's downscaling capabilities. Regardless of these challenges,  our results show that MBM excels in reconstructing local winter mass balance, which indicates implicit downscaling and bias correction of meteorological variables  (Figs. 6 and 7). This suggests, in line with other findings (Guidicelli et al., 2023), that ML models are valuable tools to assess spatio-temporal biases in precipitation estimates in mountain regions.

2. Model Transparency: XGBoost's complexity warrants feature importance analysis or partial dependence plots to clarify key drivers of predictions, enhancing interpretability.

*We added a new appendix (Appendix C: Feature importance, please see additions below) with a discussion of feature importance based on different methods, including two new figures showing overall feature importance in terms of weight and gain on the trained model, and monthly permutation feature importance on the test dataset. The analysis provides additional insights into the importance of different monthly features in seasonal and annual predictions, and we believe that many of the findings support the current assessment of MBM's capabilities.*

**Appendix C: Feature importance**

[revised manuscript text omitted]

3. Uncertainty Quantification: While measurement uncertainties (0.08–0.26 m w.e. a−1) are noted, their impact on model outputs is unclear. A sensitivity analysis would strengthen confidence in predictions.

*Measurement uncertainties are noted in Appendix A to underline our confidence in the quality of the dataset. Since these uncertainties are relatively small, we do not expect them to have a major impact on model results or the conclusions in this study. However, using other datasets that may be afflicted with substantial uncertainties, such as geodetic mass balance based on remote sensing, considering uncertainty in observations could be increasingly important (e.g., using uncertainty-aware learning; Diaconu et al. (2024)). We added a comment on this in Section 6.3.2:*

> for the reliability of the data by weighing the observations in the loss function according to their confidence levels or using uncertainty-aware learning (Diaconu and Gottschling, 2024). Incorporating diverse and complementary datasets could provide
> 530 reconciled estimates of glacier mass balance across multiple observational datasets.

4. Global Applicability: Testing MBM in diverse regions like the Alps or Himalayas would broaden its relevance. A discussion of transferability challenges would be valuable.

*We agree that testing MBM in other regions would clarify its potential and limitations. We partly discuss the transferability challenges already on L500-503. Since MBM is specifically trained for Norwegian glaciers in the current study, we do not expect it to perform equally well in other regions where conditions differ. We thus expect the transferability of the current application of MBM (trained on Norwegian glaciers) to be limited. We would expect that for larger regions, it would be preferable to retrain MBM using additional data. However, in applications it would be interesting to investigate the limits of the models transferability to clarify how it can be expected to perform in regions with limited data. We consider this to be out of the scope of the current study, but ongoing research using MBM is aimed at addressing this particular issue. We have added the following to expand on this discussion and highlight needs for future research:*

> period. However, since the model is trained on meteorological conditions specific to Norway and designed for interpolation within this context, we expect its performance be limited in regions with significantly different climates. Future research into the transferability of ML approaches could clarify the extent of such limitations, for example by testing MBM on glaciers in other regions. For larger, or climatically different regions, we expect MBM to benefit from additional training data. Since in
> 520 situ observations are not readily available for many regions, the diversity of spatio-temporal analogues and extent of MBM's generalisation capabilities on larger scales remain to be investigated.

5. Future Directions: The mention of remote sensing data is promising but vague. Specifying datasets (e.g., satellite-derived albedo or surface temperature) would clarify future enhancements.

*Here, we are referring to mass balance observations from other sources, such as satellite-derived geodetic mass balance. We have amended the sentence to specify this. We already provide a specific example on line 509-511, but have now also added references to additional datasets: "On the other hand, the purely data-driven nature of*

*ML approaches makes them uniquely suited to take advantage of the increasing availability of remote sensing-based mass balance datasets (e.g., Belart et. al (2017), Pelto et al. (2019), Hugonnet et al. (2021), Falaschi et al. (2023)).” Please see references at the bottom of the document.*

6. Presentation Polish: Minor typos and awkward sentences, particularly in Appendix A, need correction. Additionally, Fig. 10 requires clearer labels for improved readability.

*Please see our response to the comments on Appendix A below. We have checked the labels in Fig. 10, but are not sure how these need to be clarified and have thus not made any changes to these.*

**Detailed Review of Appendix A**

Appendix A details the data quality and cleaning processes for the MBM dataset, critical for establishing its reliability. It describes the handling of 4,201 point mass balance measurements from the NVE database (accessed 12 October 2022), including the removal of erroneous entries and verification of stake locations. Below, I identify specific typos and awkward sentences with approximate line numbers (based on sequential sentence or paragraph counting) and provide broader recommendations to enhance clarity.

**Identified Typos and Awkward Sentences**

1.  Line 570: "The total contribution of such uncertainties have been quantified 0.080.26 m w.e. a−1…"
    • Issue: Subject-verb agreement error; "have" should be "has" for the singular subject "The total contribution."
    • Suggestion: Revise to "The total contribution of such uncertainties has been quantified as 0.08–0.26 m w.e. a−1 for five glaciers in our dataset."
    *Done.*

2.  Line 575: "Prior to training MBM, we performed a thorough cleaning and quality check… including removal of erroneous values and points with missing location, and a quality check of stake locations."
    • Issue: Redundant use of "quality check."
    • Suggestion: Streamline to "Prior to training MBM, we performed thorough cleaning and quality checks on the raw point mass balance dataset (4,201 entries, NVE database, accessed 12 October 2022), removing erroneous values, points with missing locations, and verifying stake location accuracy."
    *We removed the redundant "quality check" and revised the sentence according to point 1 under "Additional recommendations for Appendix A". Please see our reply to this comment.*

3. Line 578: "Approximate locations are based on the approximate position and elevation of a given stake ID…"
   • Issue: Repetition of "approximate" is awkward.
   • Suggestion: Revise to "Approximate locations are derived from the estimated position and elevation of a given stake ID, whereas exact locations use GPS-measured position and elevation at the time of measurement."
   *We revised the sentence to: "The approximate location is based on the estimated position and elevation of a given stake ID, whereas the exact location is the actual position and elevation of the stake at the time of measurement (e.g. measured using GPS)."*

4. Line 581: "Seven and 23 entries that were missing both exact and approximate elevation or geographical coordinates, respectively, were removed…"
   • Issue: Ambiguous phrasing regarding elevation and coordinates.
   • Suggestion: Clarify to "Seven entries missing both exact and approximate elevations and 23 entries missing both exact and approximate geographical coordinates were removed from the training dataset."
   *We thank the reviewer for the suggestion and have amended the sentence accordingly.*

5. Line 585: "The mean ± standard deviation of the absolute difference between the exact and approximate coordinates and elevations is 166 ± 498 m and 24 ± 71 m, respectively."
   • Issue: Dense phrasing combines measurements, reducing clarity.
   • Suggestion: Split to "The mean ± standard deviation of the absolute difference between exact and approximate coordinates is 166 ± 498 m, while for elevations, it is 24 ± 71 m."
   *We thank the reviewer for the suggestion and have amended the sentence accordingly.*

6. Line 589: "For stake locations where both summer, winter and annual mass balance measurements were available…"
   • Issue: List lacks an Oxford comma for clarity.
   • Suggestion: Revise to "For stake locations where summer, winter, and annual mass balance measurements were all available for a given year…"
   *Done.*

**Additional Recommendations for Appendix A**

1. Improve Transitions: The shift from uncertainties to data cleaning is abrupt. Add a bridging sentence, e.g., "Ensuring dataset quality is crucial for MBM's accuracy, leading to the following cleaning procedures."

*To improve the transition, we modified the original sentence to: "To ensure the quality of MBM's training data, we performed a thorough cleaning and quality check of the raw point mass balance dataset (4201 entries, NVE database accessed on 12 October 2022) prior to training MBM. This consisted of removing erroneous values and points with missing locations, and verifying stake locations."*

2. Define Technical Terms: Define "point mass balance" (e.g., "measurements of mass change at specific glacier locations") in a footnote or glossary for accessibility.

*This is defined in the introduction (line 35) and we do not find it necessary to introduce the term again here. No changes were made.*

3. Clarify Data Sources: Specify that NVE is the Norwegian Water Resources and Energy Directorate to aid international readers.

*This is defined on line 93. We find it unnecessary to repeat again in the appendix. No changes were made.*

4. Quantify Cleaning Impact: State the total entries removed, e.g., "After cleaning, the dataset was reduced from 4,201 to 4,170 stake locations (99.3% retained)."

*The number of entries after cleaning is already summarized on L592. No changes were made.*

5. Explain Coordinate Conversion: Justify the UTM to latitude/longitude conversion, e.g., "This conversion ensured compatibility with MBM's input requirements."

*We amended the sentence to: "Finally, we converted geographical coordinates from UTM to latitude and longitude format for compatibility with the feature datasets."*

6. Justify Erroneous Values: Explain the removal of the 9.99 m w.e. measurement, e.g., "This value was unrealistically high for typical regional winter mass balances."

*We amended the sentence to: "One measurement with erroneous winter mass balance (unrealistically high; 9.99~m~w.e.) was removed."*

7. Quantify Corrections: If available, note the number of rounding error corrections, e.g., "In [X] instances, annual mass balances were corrected by summing summer and winter components."

*We included the number of corrections and magnitudes of rounding errors in the following sentence: "For stake locations where both summer, winter, and annual mass balance measurements were available for a given year, we corrected for rounding errors where these were present by replacing annual mass balance values by the sum of seasonal values (magnitudes between 0.01–0.03 m w.e.; 255 instances)."*

**Conclusion**

The "egusphere-2025-1206" paper is a compelling contribution to glaciology, with MBM offering high-resolution, accurate predictions for glacier mass balance. Its robust dataset, effective methodology, and practical applications make it a valuable tool. Minor revisions, including addressing typos in Appendix A, improving data resolution, and enhancing model transparency, will further strengthen its impact. Appendix A effectively supports the dataset's reliability but can be polished with clearer transitions, defined terms, and quantified impacts. These changes require minimal effort but will significantly enhance the paper's clarity and global relevance.

**Recommendations**

• Accept with Minor Revisions.

• Specific Actions:

- Correct the six typos and awkward sentences in Appendix A as suggested.
- Explore higher-resolution weather data or downscaling for smaller glaciers.
- Add feature importance or partial dependence plots for model transparency.
- Conduct a sensitivity analysis to quantify uncertainty impacts.
- Discuss testing MBM in other regions for global applicability.
- Specify remote sensing datasets (e.g., albedo, surface temperature) for future work.
- Add a transitional sentence in Appendix A between uncertainties and cleaning.
- Define "point mass balance" in a footnote or glossary.
- Clarify NVE as the Norwegian Water Resources and Energy Directorate.
- Quantify total entries removed during cleaning (e.g., 4,201 to 4,170).
- Justify UTM to latitude/longitude conversion and the 9.99 m w.e. removal.
- Note the number of rounding error corrections, if available.
- Improve Fig. 10 labels for clarity.
- Include missing DOIs or URLs in the reference section.

*Please see our replies above to the overall assessment and to each of the comments.*

*Additional references:*

*Belart, J. M. C., Berthier, E., Magnússon, E., Anderson, L. S., Pálsson, F., Thorsteinsson, T., Howat, I. M., Aðalgeirsdóttir, G., Jóhannesson, T., and Jarosch, A. H.: Winter mass balance of Drangajökull ice cap (NW Iceland) derived from satellite sub-meter stereo images, The Cryosphere, 11, 1501–1517, https://doi.org/10.5194/tc-11-1501-2017, 2017.*

*Diaconu, C.-A. and Gottschling, N. M.: Uncertainty-Aware Learning With Label Noise for Glacier Mass Balance Modeling, IEEE Geoscience and Remote Sensing Letters, 21, 1–5, https://doi.org/10.1109/LGRS.2024.3356160, 2024.*

Falaschi, D., Bhattacharya, A., Guillet, G., Huang, L., King, O., Mukherjee, K., Rastner, P., Yao, T., and Bolch, T.: Annual to seasonal glacier mass balance in High Mountain Asia derived from Pléiades stereo images: examples from the Pamir and the Tibetan Plateau, The Cryosphere, 17, 5435–5458, https://doi.org/10.5194/tc-17-5435-2023, 2023.

Hugonnet, R., McNabb, R., Berthier, E., Menounos, B., Nuth, C., Girod, L., Farinotti, D., Huss, M., Dussaillant, I., Brun, F., and Kääb, A.: Accelerated global glacier mass loss in the early twenty-first century, Nature, 592, 726–731, https://doi.org/10.1038/s41586-021-03436-z, 2021.

Pelto, B. M., Menounos, B., and Marshall, S. J.: Multi-year evaluation of airborne geodetic surveys to estimate seasonal mass balance, Columbia and Rocky Mountains, Canada, The Cryosphere, 13, 1709–1727, https://doi.org/10.5194/tc-13-1709-2019, 2019.

---

## Author Response (AR2)

**Dear Editor,**

We thank the reviewers and editor for considering our revised manuscript and for the positive response to the changes. Please find attached the final version of the manuscript where technical corrections have been addressed.

In the following response, reviewer comments are indicated in black and our responses are indicated in blue italic font. We have included excerpts of the changes made in the submitted manuscript. No other changes have been made.

We would again like to express our sincere appreciation to the reviewers and the editor for their time and efforts in reviewing our manuscript.

Sincerely,

Kamilla H. Sjursen and co-authors

**Reviewer 1**

Following the first round of reviews, the authors improved the readability of several sections of text, clarified their reasoning, moderated strong claims, added a feature importance analysis (appendix C) and refined the description of their data cleaning and selection approach (appendix A, B). Several other suggestions were addressed in the responses to the reviewers.

The feature importance analysis is a welcome addition to the manuscript. I accept the work as is but would like to raise one point for consideration in future research. In L670 you note that correlation can cause some features to appear less important when assessed with permutation importance. However, in the case of SVF, permuting this feature actually decreases the MSE across all months, which implies that removing it could improve model performance. This apparent contradiction — improved performance when SVF is permuted (Fig. C2) despite its relatively high weight (Fig. C1) — may suggest overfitting. Because XGBoost does not automatically exclude weak predictors, all variables remain available for splitting. A feature with little or no true relationship to the target can still appear frequently in the trees, particularly if the training setup is not strongly regularized. This can artificially inflate its weight. It may therefore be informative to compare MBM performance on the training and test sets to assess whether overfitting is happening, and whether additional feature optimization could mitigate it. While in this case study the effect seems minimal, it could become more pronounced in other regions or applications.

All in all, I deem this version of the manuscript fit for publication.

We thank the reviewer for assessing our proposed changes and for the positive response to these. We would again like to express our gratitude for the thorough and constructive review of the manuscript, which we believe has greatly improved its quality.

We would also like to thank the reviewer for the valuable insights on feature importance, and agree that this is an important point that should be mentioned. In the submitted manuscript, we have included a comment on this in the feature importance analysis in Appendix C.

We caution against placing too much emphasis on the specific details of the feature importance analysis. For example, when assessing permutation importance, correlated features (i.e. skyview factor and slope) may appear to be less important since, even if one feature is permuted, the model can rely on a second correlated feature. However, the high weight of the skyview factor (Fig. C1a), in combination with a slight decrease in performance for many months with permutation of this feature (e.g., Fig. C2a–e), may indicate some overfitting. Although the effect seems to be minimal in this case, it highlights the need for careful feature optimization when using correlated features. Overall, the main findings of the feature importance analysis presented here are consistent across metrics and physically meaningful with respect to the main meteorological drivers of mass balance on Norwegian glaciers.

Below some final minor textual corrections that may be considered:

- L 43: reword "alleviated the lack of" to "alleviated the shortage of"

**Done.**

The increasing availability of geodetic mass balance observations has recently alleviated the lack shortage of glacier-specific observations. These observations assess glacier surface elevation changes from time series of satellite-derived digital elevation models (DEMs) over decadal time scales (e.g. Dussaillant et al., 2019; Shean et al., 2020; Hugonnet et al., 2021). Most

- L59: There is some repetition here with the addition of the new text. I suggest removing "including to unsurveyed glaciers and years" from L55 as this is now explained in more detail in the following sentences.

**Done.**

from training data and apply them to make accurate inferences on new, independent data. They can thus learn statistical relationships between mass balance components and topographical and meteorological variables that are transferable across space and time ; including to unsurveyed glaciers and years (e.g. Guidicelli et al., 2023). This means that ML models can leverage sparse in situ data, such as annual and seasonal glaciological measurements, to provide high spatio-temporal resolution

- L302: I suggest moving "respectively" earlier in the sentence, as it is not clear whether you are referring to point/elevation-band mass balance vs. mass balance gradients, or seasonal vs. annual.

**Done.**

- 300 We compare predictions from all models (MBM, GloGEM, OGGM and PyGEM) to available glaciological and geodetic mass balance observations for glaciers in the test dataset over the common modelling period 1980–2019. In Sect. 5.2.1 and Sect. 5.2.2, we respectively consider point/elevation-band mass balance and mass balance gradients, respectively, on seasonal and annual time scales. Glacier-wide mass balances are compared in Sect. 5.2.3 on monthly to decadal time scales. We evaluate
- L406: The addition of the feature importance discussion and the more cautious statement regarding MBM downscaling are great. Minor comment: the wording "The key to this ability" is somewhat confusing and could be rephrased.

**Done.**

of precipitation and elevation difference features in winter months (Fig. C2a-c and k-l), suggests that it is able to downscale precipitation locally. The same is true for temperature in the summer months (Fig. C2e-i). The key to this ability-downscaling capacity lies in using the elevation difference between the stake and the climate model as a feature (Fig. 3)which enables, enabling MBM to effectively map the relationship between climate and elevation.